# Exploit Gradient Skew to Circumvent Byzantine Defenses for Federated Learning

## Abstract

Federated Learning (FL) is notorious for its vulnerability to Byzantine attacks. Most current Byzantine defenses share a common inductive bias: among all the gradients, the majorities are more likely to be honest. However, such a bias is a poison to Byzantine robustness due to a newly discovered phenomenon in this paper – gradient skew. We discover that the substantial honest gradients skew away from the optimal gradient (the average of honest gradients) as a result of heterogeneous data distribution. This gradient skew phenomenon allows Byzantine gradients to hide within the skewed honest gradients and thus be recognized as the majority. As a result, Byzantine defenses are deceived into perceiving Byzantine gradients as honest. Motivated by this observation, we propose a novel skew-aware attack called STRIKE: first, we search for the skewed honest gradients; then, we construct Byzantine gradients within the skewed honest gradients. Experiments on three benchmark datasets validate the effectiveness of our attack.

## 1 Introduction

Federated Learning (FL) (McMahan et al., 2017; Li et al., 2020) emerged as a privacy-aware learning paradigm, in which data owners, i.e., clients, repeatedly use their private data to compute local gradients and upload them to a central server. The central server collects the uploaded gradients from clients and aggregates these gradients to update the global model. In this way, clients can collaborate to train a model without exposing their private data.

Unfortunately, FL is susceptible to Byzantine attacks due to its distributed nature (Blanchard et al., 2017; Guerraoui et al., 2018). A malicious party can control a small subset of clients, i.e., Byzantine clients, to degrade the utility of the global model. During the training phase, Byzantine clients can send arbitrary messages to the central server to bias the global model. A wealth of defenses (Blanchard et al., 2017; Pillutla et al., 2019; Shejwalkar & Houmansadr, 2021) have been proposed to defend against Byzantine attacks in FL. They aim to estimate the optimal gradient, i.e., the average of gradients from honest clients, in the presence of Byzantine clients.

Most existing defenses (Blanchard et al., 2017; Shejwalkar & Houmansadr, 2021; Karimireddy et al., 2022) share a common inductive bias: the majority gradients are more likely to be honest. Generally, they assign higher weights to the majority gradients. Then they compute the global gradient and use it to update the global model. As a result, the output global gradient of defenses is biased towards the majority of gradients.

However, this inductive bias of Byzantine defenses is harmful to Byzantine robustness in FL due to the presence of gradient skew. In practical FL, data across different clients is non-independent and identically distributed (non-IID), which gives rise to heterogeneous honest gradients (McMahan et al., 2017; Li et al., 2020; Karimireddy et al., 2022). On closer inspection, we find that these heterogenous honest gradients are highly skewed. In Figure 1, we use Locally Linear Embedding (LLE) (Roweis & Saul, 2000) to visualize the honest gradients on CIFAR-10 dataset (Krizhevsky et al., 2009) when data is non-IID split. Detailed setups and more results are provided in Appendix A. As shown in Figure 1, substantial honest gradients skew away from the optimal gradient. We term this phenomenon as "gradient skew". When honest gradients are skewed, the defenses' bias towards majority gradients is a poison to Byzantine robustness.

In fact, we can hide Byzantine gradients within the skewed majority of honest gradients as shown in Figure 1. In this case, the bias of defenses would drive the global gradient close to the skewed honest gradients but far from the optimal gradient.

In this paper, we study how to exploit the gradient skew in the more practical non-IID setting to circumvent Byzantine defenses. We first formulate the definition of gradient skew and theoretically analyze the vulnerability of Byzantine defenses under the skew. Based on the above analysis, we design a novel two-Stage aTtack based on gRadIent sKEw called STRIKE. In particular, STRIKE hides Byzantine gradients within the skewed honest gradients as shown in Figure 1. STRIKE can take advantage of the gradient skew in FL to break Byzantine defenses.

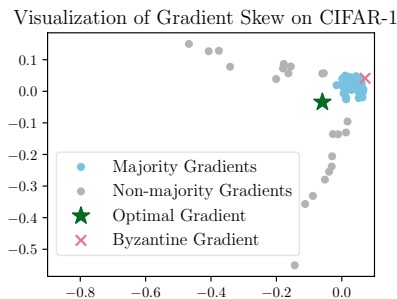

Figure 1: The LLE visualization of honest gradients in the non-IID setting on CIFAR-10. Substantial honest gradients (blue circles) are skewed away from the optimal gradient (green star). In this case, we can hide Byzantine gradients (pink crosses) within the skewed honest gradients to circumvent defenses.

In summary, our contributions are:

- To the best of our knowledge, we are the first to discover the gradient skew phenomenon in FL: substantial honest gradients are skewed away from the optimal gradient. We theoretically analyze the vulnerability of Byzantine defenses under gradient skew. In particular, we can circumvent defenses by hiding Byzantine gradients within the skewed honest gradients.

- Based on the theoretical analysis, we propose a two-stage Byzantine attack called STRIKE. In the first stage, STRIKE searches for the skewed honest gradients under the guidance of Karl Pearson's formula. In the second stage, STRIKE constructs the Byzantine gradients within the skewed honest gradients by solving a constrained optimization problem.

- Experiments on three benchmark datasets validate the effectiveness of the proposed attack. For instance, STRIKE attack improves upon the best baseline by 57.84% against DnC on FEMNIST dataset when there are 20% Byzantine clients.

## 2 RELATED WORKS

**Byzantine attacks.** Blanchard et al. (2017) first disclose the Byzantine vulnerability of FL. Baruch et al. (2019) observe that the variance of honest gradients is high enough for Byzantine clients to compromise Byzantine defenses. Based on this observation, they propose a LIE attack that hides Byzantine gradients within the variance. Xie et al. (2020) further utilize the high variance and propose an IPM attack. Particularly, they show that when the variance of honest gradients is large enough, IPM can make the inner product between the aggregated gradient and the honest average negative. However, this result is restricted to a few defenses, i.e., Median Yin et al. (2018), Trmean Yin et al. (2018), and Krum Blanchard et al. (2017). Fang et al. (2020) establish an omniscient attack called Fang. However, the Fang attack requires knowledge of the Byzantine defense, which is unrealistic in practice. Shejwalkar & Houmansadr (2021) propose Min-Max and Min-Sum attacks that solve a constrained optimization problem to determine Byzantine gradients. From a high level, both Min-Max and Min-Sum aim to maximize the perturbation to a reference benign gradient while ensuring the Byzantine gradients lie within the variance. Karimireddy et al. (2022) propose a Mimic attack that takes advantage of data heterogeneity in FL. In particular, Byzantine clients pick an honest client to mimic and copy its gradient. The above attacks take advantage of the large variance of honest gradients to break Byzantine defenses. However, they all ignore the skew nature of honest gradients in FL and fail to exploit this vulnerability.

**Byzantine resilience.** El-Mhamdi et al. (2021); Karimireddy et al. (2022) provide state-of-the-art theoretical analysis of Byzantine resilience under data heterogeneity. El-Mhamdi et al. (2021) discuss Byzantine resilience in a decentralized, asynchronous setting. Farhadkhani et al. (2022) provide a unified framework for Byzantine resilience analysis, which enables comparison among different

defenses on a common theoretical ground. Karimireddy et al. (2022) improve the error bound of Byzantine resilience to be upper-bounded by the fraction of Byzantine clients, which recovers the standard convergence rate when there are no Byzantine clients. They all share a common bias: the majority of gradients are more likely to be honest. However, this bias is a poison to Byzantine robustness in the presence of gradient skew. In practical FL, the distribution of honest gradients is highly skewed due to data heterogeneity. Therefore, existing defenses are especially vulnerable to attacks that are aware of gradient skew.

## 3 NOTATIONS AND PRELIMINARY

### 3.1 NOTATIONS

$\|\cdot\|$ denotes the $\ell_2$ norm of a vector. For vector $\boldsymbol{v}$, $(\boldsymbol{v})_k$ represents the $k$-th coordinate of $\boldsymbol{v}$. Model parameters are denoted by $\boldsymbol{w}$ and gradients are denoted by $\boldsymbol{g}$. We use $\bar{\boldsymbol{g}}$ to denote the optimal gradient, i.e., the average of honest gradients, and $\hat{\boldsymbol{g}}$ denotes the global gradients obtained by Byzantine defenses. We use subscript $i$ to denote client $i$ and use superscript $t$ to denote communication round $t$.

### 3.2 PRELIMINARY

**Federated learning.** Suppose that there are $n$ clients and a central server. The goal is to optimize the global loss function $\mathcal{L}(\cdot)$:

$$\min_{\boldsymbol{w}} \mathcal{L}(\boldsymbol{w}), \quad \text{where } \mathcal{L}(\boldsymbol{w}) = \frac{1}{n} \sum_{i=1}^{n} \mathcal{L}_i(\boldsymbol{w}). \tag{1}$$

Here $\boldsymbol{w}$ is the model parameter, and $\mathcal{L}_i(\cdot)$ is the local loss function on client $i$ for $i = 1, \ldots, n$.

In communication round $t$, the central server distributes global parameter $\boldsymbol{w}^t$ to the clients. Each client $i$ performs several epochs of SGD to minimize its local loss function $\mathcal{L}_i(\cdot)$ and update its local parameter to $\boldsymbol{w}_i^{t+1}$. Then, each client $i$ computes its local gradient $\boldsymbol{g}_i^t$ and sends it to the server.

$$\boldsymbol{g}_i^t = \boldsymbol{w}_i^t - \boldsymbol{w}_i^{t+1}, \quad i = 1, \ldots, n. \tag{2}$$

After receiving the uploaded local gradients, the server aggregates the local gradients and updates the global model to $\boldsymbol{w}^{t+1}$.

$$\bar{\boldsymbol{g}}^t = \frac{1}{n} \sum_{i=1}^{n} \boldsymbol{g}_i^t, \quad \boldsymbol{w}^{t+1} = \boldsymbol{w}^t - \bar{\boldsymbol{g}}^t. \tag{3}$$

**Byzantine attack model.** Assume that among the total $n$ clients, $f$ fixed clients are Byzantine clients. Let $\mathcal{B} \subseteq \{1, \ldots, n\}$ denote the set of Byzantine clients and $\mathcal{H} = \{1, \ldots, n\} \setminus \mathcal{B}$ denote the set of honest clients. In each communication round, Byzantine clients can send arbitrary messages to bias the global model. The local gradients that the server receives in the $t$-th communication round are

$$\boldsymbol{g}_i^t = \begin{cases} *, & i \in \mathcal{B}, \\ \boldsymbol{w}^t - \boldsymbol{w}_i^{t+1}, & i \in \mathcal{H}, \end{cases} \tag{4}$$

where $*$ represents an arbitrary message. Following Baruch et al. (2019); Xie et al. (2020), we consider the setting where the attacker only has the knowledge of honest gradients.

**Byzantine resilience.** Blanchard et al. (2017) show that the popular mean aggregation rule is not resilient to Byzantine attacks. Thus, the server replaces the mean aggregation rule in Equation (3) with a robust AGgregation Rules (AGR) $\mathcal{A}$, e.g., Krum (Blanchard et al., 2017), Median (Yin et al., 2018), to compute the global gradient $\hat{\boldsymbol{g}}^t$ and update the global model to $\boldsymbol{w}^{t+1}$.

$$\hat{\boldsymbol{g}}^t = \mathcal{A}(\boldsymbol{g}_1^t, \ldots, \boldsymbol{g}_n^t), \quad \boldsymbol{w}^{t+1} = \boldsymbol{w}^t - \hat{\boldsymbol{g}}^t. \tag{5}$$

A body of recent works (Farhadkhani et al., 2022; Karimireddy et al., 2022; Allouah et al., 2023) have theoretically defined Byzantine resilience for general robust AGRs. Particularly, we adopt the definition from Farhadkhani et al. (2022) in this work for analysis. We also discuss how our analysis can apply to other definitions of Byzantine resilience in Appendix B.2.

**Definition 1** (($f, \lambda$)-resilient). Given $f < n$ and $\lambda \geq 0$, an AGR $\mathcal{A}$ is ($f, \lambda$)-resilient if for any collection of $n$ vectors $\{\boldsymbol{g}_1, \ldots, \boldsymbol{g}_n\}$ and any set $\mathcal{G} \subseteq \{1, \ldots, n\}$ of size $n - f$,

$$\|\mathcal{A}(\boldsymbol{g}_1, \ldots, \boldsymbol{g}_n) - \bar{\boldsymbol{g}}_{\mathcal{G}}\| \leq \lambda \max_{i,j \in \mathcal{G}} \|\boldsymbol{g}_i - \boldsymbol{g}_j\|, \tag{6}$$

where $\bar{\boldsymbol{g}}_{\mathcal{G}} = \sum_{i \in \mathcal{G}} \boldsymbol{g}_i / (n - f)$ is the average of gradients $\{\boldsymbol{g}_i \mid i \in \mathcal{G}\}$.

Essentially, a smaller $\lambda$ means better resilience (Farhadkhani et al., 2022).

## 4 VULNERABILITY OF ROBUST AGRs UNDER GRADIENT SKEW

In this section, we show that when honest gradients are skewed, we can establish Byzantine attacks to circumvent robust AGgregation Rules (AGRs). First, we verify the existence of gradient skew in FL and formally define gradient skew. Then, we show how to exploit the gradient skew to launch Byzantine attacks and circumvent robust AGRs.

### 4.1 GRADIENT SKEW IN FL DUE TO NON-IID DATA

Plenty of works (Baruch et al., 2019; Xie et al., 2020; Karimireddy et al., 2022) have explored how large variance can be harmful to Byzantine robustness. However, to the best of our knowledge, none of the existing works is aware of the skewed nature of honest gradients in the non-IID setting and how gradient skew can threaten Byzantine robustness.

We take a close look at the distribution of honest gradients in the non-IID setting (without attack). To construct our FL setup, we split CIFAR-10 (Krizhevsky et al., 2009) dataset in a non-IID manner among 100 clients. For more setup details, please refer to Appendix A.1. We run FedAvg (McMahan et al., 2017) for 200 communication rounds. We randomly sample a communication round and use Locally Linear Embedding (LLE) (Roweis & Saul, 2000) to visualize the gradients in this communication round in Figure 1. From Figure 1, we observe that the majority of honest gradients (blue circles) skew away from the optimal gradient (green stars). More visualization results can be found in Appendix A.2. We name this phenomenon "gradient skew".

We formulate the definition of gradient skew for further analysis. The idea behind this definition is to measure the skewness of honest gradients by the distance between the majority of honest gradients and the optimal gradient, i.e., the average of honest gradients.

**Definition 2** (($f, \gamma$)-skewed). The set of honest gradients $\{\boldsymbol{g}_i \mid i \in \mathcal{H}\}$ is called ($f, \gamma$)-skewed if there exists a set $\mathcal{S} \subseteq \mathcal{H}$ of size $n - 2f$[1] such that

$$\mathbb{E}\left[\|\bar{\boldsymbol{g}}_{\mathcal{S}} - \bar{\boldsymbol{g}}\|^2\right] \geq \gamma \rho_{\mathcal{S}}^2, \tag{7}$$

where $\bar{\boldsymbol{g}} = \sum_{i \in \mathcal{H}} \boldsymbol{g}_i / (n - f)$, $\bar{\boldsymbol{g}}_{\mathcal{S}} = \sum_{i \in \mathcal{S}} \boldsymbol{g}_i / (n - 2f)$, and $\rho_{\mathcal{S}}^2 = \mathbb{E}\left[\max_{i,j \in \mathcal{S}} \|\boldsymbol{g}_i - \boldsymbol{g}_j\|^2\right]$ is a measure of gradient heterogeneity introduced by El-Mhamdi et al. (2021). Here, gradients $\{\boldsymbol{g}_i \mid i \in \mathcal{S}\}$ are called the *skewed honest gradients*, and $\gamma$ is called the skewness of honest gradients $\{\boldsymbol{g}_i \mid i \in \mathcal{H}\}$.

In Definition 2, $\gamma$ measures the skew degree of the honest gradients. A larger $\gamma$ indicates a higher skew degree. We provide an example in Appendix B to help better understand the gradient skew.

### 4.2 ROBUST AGRs ARE BRITTLE UNDER GRADIENT SKEW

When the honest gradients are skewed, robust AGRs are extremely vulnerable. In fact, we can hide Byzantine gradients within the skewed honest gradients. This attack strategy makes Byzantine gradients stealthy and difficult to detect. The skewed nature of honest gradients further allows Byzantine gradients to deviate the global gradient away from the optimal gradient. The above argument can be formulated as the following lower bound.

---

[1] The size of the skewed honest gradients $|\mathcal{S}| = n - 2f$ is carefully chosen as follows: We aim to deceive robust AGR into believing that Byzantine gradients and skewed honest gradients are honest, and the other honest gradients are Byzantine. There are $n - f$ honest gradients and $f$ Byzantine gradients. Thus the number of skewed honest gradients is determined by $|\mathcal{S}| = (n - f) - f = n - 2f$.

**Proposition 1** (Vulnerability under skew). *Given any $(f, \lambda)$-resilient AGR $\mathcal{A}$, $\gamma = \Omega(\lambda^2)$, if the set of honest gradients $\{\boldsymbol{g}_i \mid i \in \mathcal{H}\}$ is $(f, \gamma)$-skewed, then there exist Byzantine gradients $\{\boldsymbol{g}_i \mid i \in \mathcal{B}\}$ such that*

$$\mathbb{E}[\|\mathcal{A}(\boldsymbol{g}_1, \ldots, \boldsymbol{g}_n) - \bar{\boldsymbol{g}}\|^2] \geq \Omega(\frac{\gamma}{\lambda^2} \cdot \frac{f^2}{(n-f)^2} \cdot \rho_{\mathcal{S}}^2). \tag{8}$$

*where $\bar{\boldsymbol{g}} = \sum_{i \in \mathcal{H}} \boldsymbol{g}_i / (n - f)$ is the optimal gradient, $\rho_{\mathcal{S}}^2 = \mathbb{E}[\max_{i,j \in \mathcal{S}} \|\boldsymbol{g}_i - \boldsymbol{g}_j\|^2]$, $\mathcal{S}$ is the index set of the skewed honest gradients.*

The detailed proof is provided in Appendix B.1.2. Proposition 1 suggests that when the honest gradients are skewed, we can always launch Byzantine attacks to deviate the global gradient from the optimal gradient. Moreover, the more skewed the honest gradients are, the farther the global gradient is from the optimal gradient. An interesting result in Proposition 1 is that smaller $\lambda$ leads to a larger lower bound in Equation (8), which implies that our attack is even more effective on robust AGRs with stronger resilience. This is because the global gradient obtained by robust AGRs with stronger resilience is closer to the majority of uploaded gradients (including Byzantine and honest). And the majority of uploaded gradients are away from the optimal gradients under our attack. Therefore, a robust AGR with stronger resilience is even more sensitive to our attack.

We further show that the above vulnerability enables us to prevent the global model from converging to the optimum for any $L$-smooth global loss function and unbiased honest gradients. These assumptions are standard in Byzantine robust learning (Karimireddy et al., 2021; Farhadkhani et al., 2022).

**Assumption 1** ($L$-smooth). The loss function is $L$-smooth, i.e.,

$$\|\nabla \mathcal{L}(\boldsymbol{w}) - \nabla \mathcal{L}(\boldsymbol{w}')\| \leq L\|\boldsymbol{w} - \boldsymbol{w}'\|, \quad \forall \boldsymbol{w}, \boldsymbol{w}' \in \mathbb{R}^d. \tag{9}$$

**Assumption 2** (Unbias). The stochastic gradients sampled from any local data distribution are unbiased estimators of local gradients for all clients, i.e.,

$$\mathbb{E}[\boldsymbol{g}_i^t] = \nabla \mathcal{L}_i(\boldsymbol{w}^t), \quad \forall i = 1, \ldots n, t = 0, \ldots, T-1. \tag{10}$$

Now we present our main result.

**Proposition 2.** *Given any $(f, \lambda)$-resilient AGR $\mathcal{A}$, $L$-smooth global loss function $\mathcal{L}$, and learning rate $\eta \leq 1/L$, $\gamma = \Omega(\lambda^2)$, if honest gradients $\{\boldsymbol{g}_i^t \mid i \in \mathcal{H}\}$ are $(f, \gamma)$-skewed for all $t = 0, \ldots, T - 1$, then there exist Byzantine gradients $\{\boldsymbol{g}_b^t \mid b \in \mathcal{B}, t = 0, \ldots, T - 1\}$ such that the global model parameter is bounded away from the global optimum $\boldsymbol{w}^*$:*

$$\mathbb{E}[\|\boldsymbol{w}^t - \boldsymbol{w}^*\|^2] \geq \Omega(\eta^2(1 - L\eta)^2 \cdot \frac{\gamma}{\lambda^2} \cdot \frac{f^2}{(n-f)^2} \cdot \rho^2), \quad t = 1, \ldots, T, \tag{11}$$

*where $\boldsymbol{w}^t$ is the parameter of global model in the $t$-th communication round, $\boldsymbol{w}^*$ is the global optimum of global loss function $\mathcal{L}$, $\rho^2 = \min_{t=0,\ldots,T-1} \mathbb{E}[\max_{i,j \in \mathcal{S}^t} \|\boldsymbol{g}_i^t - \boldsymbol{g}_j^t\|^2]$, and $\mathcal{S}^t$ is the index set of the skewed honest gradients in $t$-th communication round.*

The proof of Proposition 2 can be found in Appendix B.1.3. Proposition 2 indicates that under gradient skew, we can establish Byzantine attacks to keep the global model away from the optimum. The lower bound in Proposition 2 is also aligned with the one in Proposition 1: a larger skewness $\gamma$ would lead to a larger lower bound, and so does a smaller $\lambda$. Note that we do not require the loss function to be convex, which implies that Proposition 2 also applies to more challenging non-convex loss functions.

## 5 PROPOSED ATTACK

In this section, we introduce the proposed two-Stage aTtack based on gRadIent sKEw called STRIKE. As discussed in Section 4, the attack principle of STRIKE is to hide Byzantine gradients within the skewed gradients. To achieve this goal, we carry out STRIKE attack in two stages: in the first stage, we search for the skewed honest gradients; in the second stage, we construct Byzantine gradients within the skewed honest gradients found in the first stage. The procedure of STRIKE attack is shown in Algorithm 1 in Appendix C.

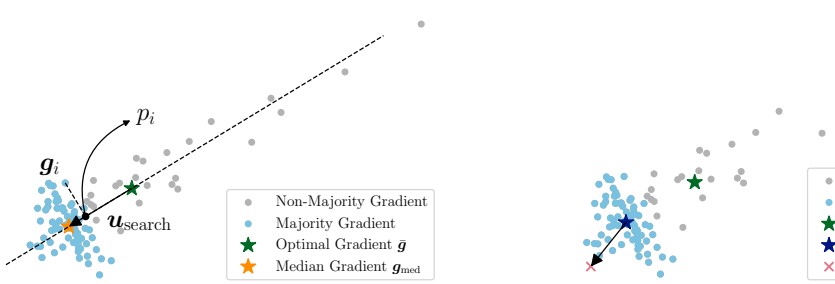

(a) We search along the direction $\boldsymbol{u}_{\text{search}} = \boldsymbol{g}_{\text{med}} - \bar{\boldsymbol{g}}$. The honest gradients with the largest scalar projection $p_i$ are selected as the skewed honest gradients (blue circles).

(b) We start from the average of skewed honest gradients $\bar{\boldsymbol{g}}_{\mathcal{S}}$ (dark blue star) and select $\alpha$ such that Byzantine gradient $\boldsymbol{g}_b$ (pink cross) lies within the skewed honest gradients.

Figure 2: Illustration of the proposed two-stage attack STRIKE: in the first stage, STRIKE searches for the skewed honest gradients; in the second stage, STRIKE hides Byzantine gradients within the skewed honest gradients.

**Search for the skewed honest gradients.** To hide the Byzantine gradient in the skewed honest gradients, we first need to find the skewed honest gradients. Naively searching the skewed honest gradients according to Definition 2 is computationally expensive. Therefore, we perform a heuristic search motivated by Karl Pearson's formula (Knoke et al., 2002; Moore et al., 2009). Figure 2a illustrates the search procedure in this stage.

As visualized in Figure 1, gradients are densely distributed within the skewed honest gradients, which implies that the population mode coincides with the skewed honest gradients with high probability. Karl Pearson's formula (Knoke et al., 2002; Moore et al., 2009) implies that the mode and median lie on the same side of the mean. Therefore, we search for the skewed honest gradients along the direction $\boldsymbol{u}_{\text{search}}$ defined as:

$$\boldsymbol{u}_{\text{search}} = \boldsymbol{g}_{\text{med}} - \bar{\boldsymbol{g}}, \tag{12}$$

where $\boldsymbol{g}_{\text{med}}$ is the coordinate-wise median of honest gradients $\{\boldsymbol{g}_i \mid i \in \mathcal{H}\}$, i.e., the $k$-th coordinate of $\boldsymbol{g}_{\text{med}}$ is $(\boldsymbol{g}_{\text{med}})_k = \text{median}\{(\boldsymbol{g}_i)_k \mid i \in \mathcal{H}\}$, and $\bar{\boldsymbol{g}} = \sum_{i \in \mathcal{H}} \boldsymbol{g}_i / (n - f)$ is the average of honest gradients.

For each honest gradient $\boldsymbol{g}_i$, we compute its scalar projection $p_i$ on the searching direction $\boldsymbol{u}_{\text{search}}$:

$$p_i = \langle \boldsymbol{g}_i, \frac{\boldsymbol{u}_{\text{search}}}{\|\boldsymbol{u}_{\text{search}}\|} \rangle, \quad \forall i \in \mathcal{H}, \tag{13}$$

where $\langle \cdot, \cdot \rangle$ represents the inner product. The $n - 2f$ gradients with the highest scalar projection values are identified as the skewed honest gradients. The goal is to have AGR consider the selected $n - 2f$ gradients as honest and the unselected $f$ gradients as Byzantine. Let $\mathcal{S}$ denote index set, that is

$$\mathcal{S} = \text{Set of } (n - 2f) \text{ indices of the gradients with the highest scalar projection } p_i, \tag{14}$$

then the skewed honest gradients are $\{\boldsymbol{g}_i \mid i \in \mathcal{S}\}$.

**Hide Byzantine gradients within the skewed honest gradients.** In this stage, we aim to hide Byzantine gradients $\{\boldsymbol{g}_i \mid i \in \mathcal{B}\}$ within the skewed honest gradients $\{\boldsymbol{g}_i \mid i \in \mathcal{S}\}$ identified in stage 1. The primary goal of our attack is to disguise Byzantine gradients and the skewed honest gradients $\{\boldsymbol{g}_i \mid i \in \mathcal{B} \cup \mathcal{S}\}$ as honest gradients. Meanwhile, the secondary goal is to maximize the attack effect, i.e., maximize the distance between these "fake" honest gradients and the optimal gradient. The hiding procedure in this stage is illustrated in Figure 2b.

According to Definition 1, robust AGRs are sensitive to the diameter of gradients. Therefore, we ensure that the Byzantine gradients lie within the diameter of the skewed honest gradients in order not to be detected.

$$\|\boldsymbol{g}_b - \boldsymbol{g}_s\| \leq \max_{i,j \in \mathcal{S}} \|\boldsymbol{g}_i - \boldsymbol{g}_j\|, \quad \forall b \in \mathcal{B}, s \in \mathcal{S}. \tag{15}$$

Meanwhile, we want to maximize the attack effect. Therefore, we need to maximize the distance between $\bar{\boldsymbol{g}}_{\mathcal{S} \cup \mathcal{B}} = \sum_{i \in \mathcal{S} \cup \mathcal{B}} \boldsymbol{g}_i / (n - f)$ and the optimal gradient.

$$\max_{\{\boldsymbol{g}_b | b \in \mathcal{B}\}} \|\bar{\boldsymbol{g}}_{\mathcal{S} \cup \mathcal{B}} - \bar{\boldsymbol{g}}\|. \tag{16}$$

In summary, our objective can be formulated as the following constrained optimization problem.

$$\max_{\{\boldsymbol{g}_b | b \in \mathcal{B}\}} \|\bar{\boldsymbol{g}}_{\mathcal{S} \cup \mathcal{B}} - \bar{\boldsymbol{g}}\| \quad \text{s.t.} \quad \begin{cases} \bar{\boldsymbol{g}}_{\mathcal{S} \cup \mathcal{B}} = \sum_{i \in \mathcal{S} \cup \mathcal{B}} \boldsymbol{g}_i / (n - f) \\ \|\boldsymbol{g}_b - \boldsymbol{g}_s\| \leq \max_{i,j \in \mathcal{S}} \|\boldsymbol{g}_i - \boldsymbol{g}_j\|, \quad \forall b \in \mathcal{B}, s \in \mathcal{S} \end{cases} \tag{17}$$

Equation (17) is too complex to be solved due to the high complexity of its feasible region. Therefore, we restrict $\{\boldsymbol{g}_b \mid b \in \mathcal{B}\}$ to the following form:

$$\boldsymbol{g}_b = \bar{\boldsymbol{g}}_{\mathcal{S}} + \alpha \cdot \mathrm{sign}(\bar{\boldsymbol{g}}_{\mathcal{S}} - \bar{\boldsymbol{g}}) \odot \boldsymbol{\sigma}_{\mathcal{S}}, \quad \forall b \in \mathcal{B}, \tag{18}$$

where $\bar{\boldsymbol{g}}_{\mathcal{S}} = \sum_{i \in \mathcal{S}} \boldsymbol{g}_i / (n - 2f)$ is the average of the skewed honest gradients, $\alpha$ is a non-negative real number that controls the attack strength, $\mathrm{sign}(\cdot)$ returns the element-wise indication of the sign of a number, $\odot$ is the element-wise multiplication, and $\boldsymbol{\sigma}_{\mathcal{S}}$ is the element-wise standard deviation of skewed honest gradients $\{\boldsymbol{g}_i \mid i \in \mathcal{S}\}$. $\bar{\boldsymbol{g}}_{\mathcal{S}}$ lies within the feasible region of Equation (17), which ensures that $\{\boldsymbol{g}_b \mid b \in \mathcal{B}\}$ are feasible when $\alpha = 0$. $\mathrm{sign}(\bar{\boldsymbol{g}}_{\mathcal{S}} - \bar{\boldsymbol{g}})$ controls the element-wise attack direction, and ensures that $\boldsymbol{g}_b$ is farther away from the optimal gradient $\bar{\boldsymbol{g}}$ under a larger $\alpha$. $\boldsymbol{\sigma}_{\mathcal{S}}$ controls the element-wise attack strength and ensures that Byzantine gradients are covert in each dimension.

With the restriction in Equation (18), Equation (17) can be simplified to the following optimization problem,

$$\max \alpha \quad \text{s.t.} \quad \|\bar{\boldsymbol{g}}_{\mathcal{S}} + \alpha \cdot \mathrm{sign}(\bar{\boldsymbol{g}}_{\mathcal{S}}) \odot \boldsymbol{\sigma}_{\mathcal{S}} - \boldsymbol{g}_s\| \leq \max_{i,j \in \mathcal{S}} \|\boldsymbol{g}_i - \boldsymbol{g}_j\|, \quad \forall s \in \mathcal{S}, \tag{19}$$

which can be easily solved by the bisection method described in Appendix D. While $\alpha$ that solves Equation (19) is theoretically provable (as shown in Appendix B.1.2, the proof of Proposition 1), we find in practice that an adjusted attack strength can further improve the effect of STRIKE. We use an additional hyperparameter $\nu(> 0)$ to control the attack strength of STRIKE. STRIKE sets $\boldsymbol{g}_b = \bar{\boldsymbol{g}}_{\mathcal{S}} + \nu\alpha \cdot \mathrm{sign}(\bar{\boldsymbol{g}}_{\mathcal{S}}) \odot \boldsymbol{\sigma}_{\mathcal{S}} - \boldsymbol{g}_i$ for all $b \in \mathcal{B}$ and uploads Byzantine gradients to the server. Higher $\nu$ implies higher attack strength. We discuss the performance of STRIKE with different $\nu$ in Appendix E.2.1.

## 6 EXPERIMENTS

### 6.1 EXPERIMENTAL SETUPS

**Datasets.** Our experiments are conducted on three real-world datasets: CIFAR-10 (Krizhevsky et al., 2009), a subset of ImageNet (Russakovsky et al., 2015) refered as ImageNet-12 (Li et al., 2021b) and FEMNIST (Caldas et al., 2018). Please refer to Appendix E.1.1 for more details about the data distribution.

**Baseline attacks.** We consider six state-of-the-art attacks: BitFlip (Allen-Zhu et al., 2020), LIE (Baruch et al., 2019), IPM (Xie et al., 2020), Min-Max (Shejwalkar & Houmansadr, 2021), Min-Sum (Shejwalkar & Houmansadr, 2021), and Mimic (Karimireddy et al., 2022). The detailed hyperparameter settings of these attacks are shown in Appendix E.1.2.

**Evaluated defenses.** We evaluate the performance of our attack on the following robust AGRs: Multi-Krum Blanchard et al. (2017), Median Yin et al. (2018), RFA Pillutla et al. (2019), Aksel Boussetta et al. (2021), CClip Karimireddy et al. (2021) DnC Shejwalkar & Houmansadr (2021), and RBTM El-Mhamdi et al. (2021). Besides, we also consider bucketing Karimireddy et al. (2022) and NNM Allouah et al. (2023), two simple yet effective schemes that adapt existing robust AGRs to the non-IID setting. The detailed hyperparameter settings of the above robust AGRs are listed in Appendix E.1.3.

More detailed setups are deferred to Appendix E.1.

Table 1: Accuracy (mean±std) under different attacks against different defenses on CIFAR-10, ImageNet-12, and FEMNIST. The best attack performance is in bold (the *lower*, the better).

| CIFAR-10 | | | | | | | |
|---|---|---|---|---|---|---|---|
| Attack | Multi-Krum | Median | RFA | Aksel | CClip | DnC | RBTM |
| BitFlip | $54.76 \pm 0.06$ | $53.73 \pm 2.05$ | $56.04 \pm 3.13$ | $51.99 \pm 2.04$ | $54.44 \pm 0.46$ | $60.81 \pm 0.56$ | $55.21 \pm 3.72$ |
| LIE | $57.89 \pm 0.22$ | $49.20 \pm 3.27$ | $53.90 \pm 5.43$ | $46.73 \pm 4.86$ | $63.11 \pm 0.43$ | $61.58 \pm 2.85$ | $58.84 \pm 0.64$ |
| IPM | $47.55 \pm 1.75$ | $51.68 \pm 1.85$ | $55.36 \pm 2.10$ | $56.85 \pm 2.07$ | $58.75 \pm 5.59$ | $62.30 \pm 3.60$ | $48.43 \pm 0.17$ |
| MinMax | $59.44 \pm 3.41$ | $57.27 \pm 0.63$ | $60.20 \pm 1.63$ | $57.17 \pm 5.50$ | $59.38 \pm 5.15$ | $62.53 \pm 2.67$ | $57.72 \pm 2.94$ |
| MinSum | $55.47 \pm 1.70$ | $52.27 \pm 0.53$ | $54.59 \pm 2.38$ | $56.43 \pm 1.74$ | $54.70 \pm 1.96$ | $61.89 \pm 1.62$ | $46.78 \pm 0.32$ |
| Mimic | $56.00 \pm 4.26$ | $52.55 \pm 0.89$ | $53.61 \pm 0.86$ | $57.19 \pm 2.50$ | $51.00 \pm 0.11$ | $62.10 \pm 5.22$ | $46.77 \pm 2.52$ |
| STRIKE (Ours) | $\mathbf{42.90} \pm 1.97$ | $\mathbf{48.29} \pm 0.40$ | $\mathbf{52.92} \pm 1.75$ | $\mathbf{38.31} \pm 0.47$ | $\mathbf{50.67} \pm 0.27$ | $\mathbf{59.16} \pm 1.84$ | $\mathbf{44.82} \pm 0.97$ |

| ImageNet-12 | | | | | | | |
|---|---|---|---|---|---|---|---|
| Attack | Multi-Krum | Median | RFA | Aksel | CClip | DnC | RBTM |
| BitFlip | $59.62 \pm 0.73$ | $58.56 \pm 4.80$ | $59.71 \pm 5.00$ | $61.64 \pm 1.98$ | $14.87 \pm 1.58$ | $59.78 \pm 1.50$ | $58.49 \pm 1.99$ |
| LIE | $62.66 \pm 0.30$ | $51.41 \pm 1.52$ | $60.99 \pm 1.22$ | $54.14 \pm 3.14$ | $16.19 \pm 3.95$ | $67.85 \pm 2.87$ | $67.12 \pm 0.39$ |
| IPM | $52.66 \pm 2.01$ | $59.20 \pm 2.44$ | $61.25 \pm 0.62$ | $59.17 \pm 1.27$ | $14.33 \pm 5.95$ | $66.31 \pm 3.60$ | $55.93 \pm 0.57$ |
| MinMax | $68.17 \pm 1.91$ | $67.76 \pm 0.07$ | $63.05 \pm 0.75$ | $59.33 \pm 3.85$ | $20.99 \pm 3.07$ | $68.05 \pm 1.59$ | $65.99 \pm 1.26$ |
| MinSum | $57.50 \pm 3.09$ | $58.78 \pm 2.10$ | $64.04 \pm 0.69$ | $67.15 \pm 0.32$ | $16.38 \pm 2.70$ | $68.69 \pm 1.18$ | $61.70 \pm 1.62$ |
| Mimic | $66.86 \pm 0.04$ | $59.39 \pm 6.07$ | $60.45 \pm 7.09$ | $58.94 \pm 1.27$ | $11.35 \pm 2.26$ | $69.07 \pm 4.69$ | $55.26 \pm 1.30$ |
| STRIKE (Ours) | $\mathbf{27.24} \pm 1.63$ | $\mathbf{42.98} \pm 1.62$ | $\mathbf{43.30} \pm 3.13$ | $\mathbf{38.11} \pm 1.02$ | $\mathbf{8.33} \pm 1.85$ | $\mathbf{53.40} \pm 4.94$ | $\mathbf{38.81} \pm 0.65$ |

| FEMNIST | | | | | | | |
|---|---|---|---|---|---|---|---|
| Attack | Multi-Krum | Median | RFA | Aksel | CClip | DnC | RBTM |
| BitFlip | $82.67 \pm 5.13$ | $71.57 \pm 3.61$ | $83.41 \pm 4.33$ | $81.42 \pm 3.45$ | $83.85 \pm 8.50$ | $83.58 \pm 5.20$ | $82.58 \pm 6.08$ |
| LIE | $68.11 \pm 6.86$ | $58.38 \pm 7.06$ | $66.19 \pm 7.93$ | $38.48 \pm 3.32$ | $73.03 \pm 3.86$ | $77.42 \pm 5.60$ | $53.35 \pm 5.17$ |
| IPM | $84.12 \pm 3.06$ | $72.60 \pm 8.42$ | $83.42 \pm 4.13$ | $78.28 \pm 7.37$ | $84.93 \pm 4.41$ | $83.03 \pm 5.02$ | $83.21 \pm 6.42$ |
| MinMax | $68.42 \pm 5.91$ | $66.44 \pm 5.88$ | $71.55 \pm 5.98$ | $34.22 \pm 4.94$ | $72.12 \pm 4.39$ | $75.40 \pm 3.78$ | $59.23 \pm 3.41$ |
| MinSum | $62.06 \pm 3.13$ | $65.46 \pm 3.66$ | $70.36 \pm 7.24$ | $44.91 \pm 3.90$ | $75.40 \pm 4.88$ | $77.11 \pm 3.61$ | $68.10 \pm 8.86$ |
| Mimic | $83.15 \pm 3.46$ | $74.00 \pm 4.79$ | $83.87 \pm 3.00$ | $79.06 \pm 7.21$ | $83.94 \pm 5.25$ | $82.22 \pm 5.40$ | $81.92 \pm 3.40$ |
| STRIKE (Ours) | $\mathbf{22.13} \pm 7.78$ | $\mathbf{55.19} \pm 3.49$ | $\mathbf{39.43} \pm 5.06$ | $\mathbf{16.58} \pm 3.63$ | $\mathbf{18.88} \pm 4.30$ | $\mathbf{17.56} \pm 5.95$ | $\mathbf{39.33} \pm 11.98$ |

## 6.2 EXPERIMENT RESULTS

**Attacking against various robust AGRs.** Table 1 demonstrates the performance of seven different attacks against seven robust AGRs on CIFAR-10, ImageNet-12, and FEMNIST datasets. From Table 1, we can observe that our STRIKE attack generally outperforms all the baseline attacks against various defenses on all datasets, verifying the efficacy of our STRIKE attack. On ImageNet-12 and FEMNIST, the improvement of STRIKE over the best baselines is more significant. We hypothesize that this is because the skew degree is higher on ImageNet-12 and FEMNIST compared to CIFAR-10. Since STRIKE exploits gradient skew to launch Byzantine attacks, it is more effective on ImageNet-12 and FEMNIST. DnC demonstrates almost the strongest resilience to previous baseline attacks. This is because these attacks fail to be aware of the skew nature of honest gradients in FL. By contrast, our STRIKE attack can take advantage of gradient skew and circumvent DnC defense. The above observations clearly validate the superiority of STRIKE.

**Attacking against robust AGRs with bucketing.** Figure 3 demonstrates the performance of seven different attacks against the bucketing scheme (Karimireddy et al., 2022) with different robust AGRs. The results demonstrate that our STRIKE attack works best against Multi-Krum, RFA, and Aksel. When attacking against DnC, Median, and RBTM, only MinSum attack is comparable to our STRIKE attack.

**Attacking against robust AGRs with NNM.** Table 2 compare the performance of STRKE attack against top-3 strongest attacks against the NNM scheme (Karimireddy et al., 2022) under the top-3 most robust robust AGRs. The results suggest that the proposed STRIKE attack still outperforms other baseline attacks against NNM.

**Imparct of $\nu$ on STRIKE attack.** We study the influence of $\nu$ on ImageNet-12 dataset. We report the test accuracy under STRIKE attack with $\nu$ in $\{0.25 * i \mid i = 1, \ldots, 8\}$ against seven different defenses on ImageNet-12. As shown in Figure 6, the performance of STRIKE is generally competitive with varying $\nu$. In most cases, simply setting $\nu = 1$ can beat almost all the attacks (except for CClip, yet we observe that the performance is low enough to make the model useless).

**The effectiveness of STRIKE attack under different non-IID levels.** We vary Dirichlet concentration parameter $\beta$ in $\{0.1, 0.2, 0.5, 0.7, 0.9\}$ to study how our attack behaves under different non-IID levels. We additionally test the performance in the IID setting. As shown in Figure 7, the

| Attack | NNM + Median | NNM + RFA | NNM + DnC |
|---|---|---|---|
| BitFlip | 57.14 | 58.55 | 53.68 |
| LIE | 58.04 | 58.68 | 58.87 |
| Mimic | 66.15 | 67.43 | 69.35 |
| STRIKE | **39.61** | **40.38** | **38.91** |

Table 2: Accuracy under different attacks against different defenses on ImageNet-12. The best results are in bold (The *lower*, the better).

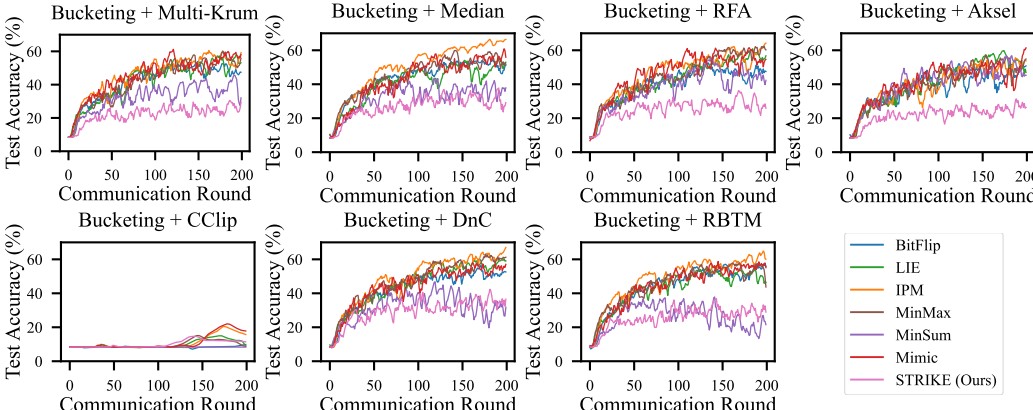

Figure 3: Accuracy under different attacks against seven robust AGRs with bucketing on ImageNet-12. The *lower*, the better.

accuracy generally increases as $\beta$ decreases for all attacks. The accuracy under our STRIKE attack is consistently lower than that of all the baseline attacks. Besides, we also note that the accuracy gap between our STRIKE attack and other baseline attacks gets smaller when the non-IID level decreases. We hypothesize the reason is that gradient skew becomes milder as the non-IID level decreases, which aligns with our theoretical results in Propositions 1 and 2. Even in the IID setting, our STRIKE attack is competitive compared to other baselines.

**The performance of STRIKE attack with different Byzantine client ratio.** We vary the number of Byzantine clients $f$ in $\{5, 10, 15, 20\}$ and fix the total number of clients $n$ to be 50. In this way, Byzantine client ratio $f/n$ varies in $\{0.1, 0.2, 0.3, 0.4\}$ to study how our attack behaves under different Byzantine client ratio. As shown in Figure 8, the accuracy generally decreases as $f/n$ increases for all attacks. The accuracy under our STRIKE attack is consistently lower than that under all the baseline attacks.

**The performance of STRIKE attack with different client number.** We vary the number of total clients $n$ in $\{10, 30, 50, 70, 90\}$ and set the number of Byzantine clients $f = 0.2n$ accordingly. The results are plotted in Figure 9 in Appendix E.2.4. As shown in Figure 9, the accuracy generally decreases as client number $n$ increases for all attacks. The accuracy under our STRIKE attack is consistently lower than that under all the baseline attacks with different number of clients.

## 7 CONCLUSION

In this paper, we theoretically analyze the vulnerability of existing defenses in the non-IID setting due to the skewed nature of honest gradients. Based on the analysis, we propose a novel attack called STRIKE that can exploit the vulnerability. Generally, STRIKE hides Byzantine gradients within the skewed honest gradients. To this end, STRIKE first searches for the skewed honest gradients, and then constructs Byzantine gradients within the skewed honest gradients by solving a constrained optimization problem. Empirical studies on three real-world datasets confirm the efficacy of our STRIKE attack. The STRIKE relies on the gradient skew phenomenon, which is closely related to non-IIDness of data distribution. When the data is IID, the performance could be limited. Therefore, defenses that can alleviate non-IID can potentially mitigate our STRIKE attack.

ETHICS AND BROADER IMPACT

The proposed skew-aware Byzantine attack STRIKE can present a threat to federated learning. Our goal with this work is thus to preempt these harms and encourage Byzantine defenses that are robust to skew-aware attacks in the future.

REPRODUCIBILITY STATEMENT

The implementation code is provided in Supplementary Materials. All datasets and the code platform (PyTorch) we use are public. Detail experiment setups are provided in the Appendix E.

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

## A   VISUALIZATION OF GRADIENT SKEW

In order to gain insight into the gradient distribution, we use Locally Linear Embedding (LLE) [2] (Roweis & Saul, 2000) to visualize the gradients. From the visualization results, we observe that the distribution of gradient is skewed throughout FL training process when the data across different clients is non-IID. In this section, we first provide the detailed experimental setups of the observation experiments and then present the visualization results.

We also visualize the Byzantine gradients together with honest gradients under STRIKE attack against Median AGR on CIFAR-10 in the non-IID setting in Figure 5. The visualization shows that Byzantine gradients can hide within the skewed honest gradients well. This justifies that the heuristic search in the first stage of STRIKE attack can effectively find the skewed honest gradients.

### A.1   EXPERIMENTAL SETUPS

For CIFAR-10, we set the number of clients $n = 100$ and the Dirichlet concentration parameter $\beta = 0.1$. For ImageNet-12, we set the number of clients $n = 50$ and the Dirichlet concentration parameter $\beta = 0.1$. For FEMNIST, we adopt its natural data partition as introduced in Section 6.1. For all three datasets, we set the number of Byzantine clients $f = 0$. For CIFAR-10 and FEMNIST, we sample 100 clients to participate in training in each communication round. More visualized gradients would help us capture the characteristic of gradient distribution. For ImageNet-12, we sample 50 clients in each communication round. This is because we train ResNet-18 on ImageNet-12 and LLE on 100 gradients of ResNet-18 would be intractable due to the high dimensionality. Other setups align with Table 5.

For LLE, we set the number of neighbors to be $k = 0.1m$, where $m$ is the number of sampled clients, to capture both local and global geometry of gradient distribution.

### A.2   GRADIENT VISUALIZATION RESULTS

On each dataset, we run FedAvg for $T$ communication round. Among the total $T$ communication rounds, we randomly sample 6 rounds for visualization. For each round, we use LLE to visualize all the gradients and the optimal gradient (the average of all gradients) in this round. Please note that LLE is not linear. Therefore, the optimal gradient after the LLE may not be the average of all uploaded gradients after LLE. The visualization results are posted in Figure 4 below. In Figure 4, the substantial gradients skew away from the optimal gradient. These results imply that the gradient distribution is skewed during the entire training process.

## B   THEORETICAL ANALYSIS: EXPLOIT GRADIENT SKEW TO CIRCUMVENT BYZANTINE DEFENSES

We first recall all the definitions and assumptions for the integrity of this section.

**Definition 1** (($f, \lambda$)-resilient). Given $f < n$ and $\lambda \geq 0$, an AGR $\mathcal{A}$ is ($f, \lambda$)-resilient if for any collection of $n$ vectors $\{\boldsymbol{g}_1, \ldots, \boldsymbol{g}_n\}$ and any set $\mathcal{G} \subseteq \{1, \ldots, n\}$ of size $n - f$,

$$\|\mathcal{A}(\boldsymbol{g}_1, \ldots, \boldsymbol{g}_n) - \bar{\boldsymbol{g}}_{\mathcal{G}}\| \leq \lambda \max_{i,j \in \mathcal{G}} \|\boldsymbol{g}_i - \boldsymbol{g}_j\|, \tag{20}$$

where $\bar{\boldsymbol{g}}_{\mathcal{G}} = \sum_{i \in \mathcal{G}} \boldsymbol{g}_i / (n - f)$ is the average of gradients $\{\boldsymbol{g}_i \mid i \in \mathcal{G}\}$.

---

[2]Compared to LLE, t-SNE (Van der Maaten & Hinton, 2008) is a more popular visualization technique. Since t-SNE adjusts Gaussian bandwidth to locally normalize the density of data points, t-SNE can not capture the distance information of data. However, gradient skew relies heavily on distance information. Therefore, t-SNE is not appropriate for the visualization of gradient skew. In contrast, LLE can preserve the distance information of data distribution.

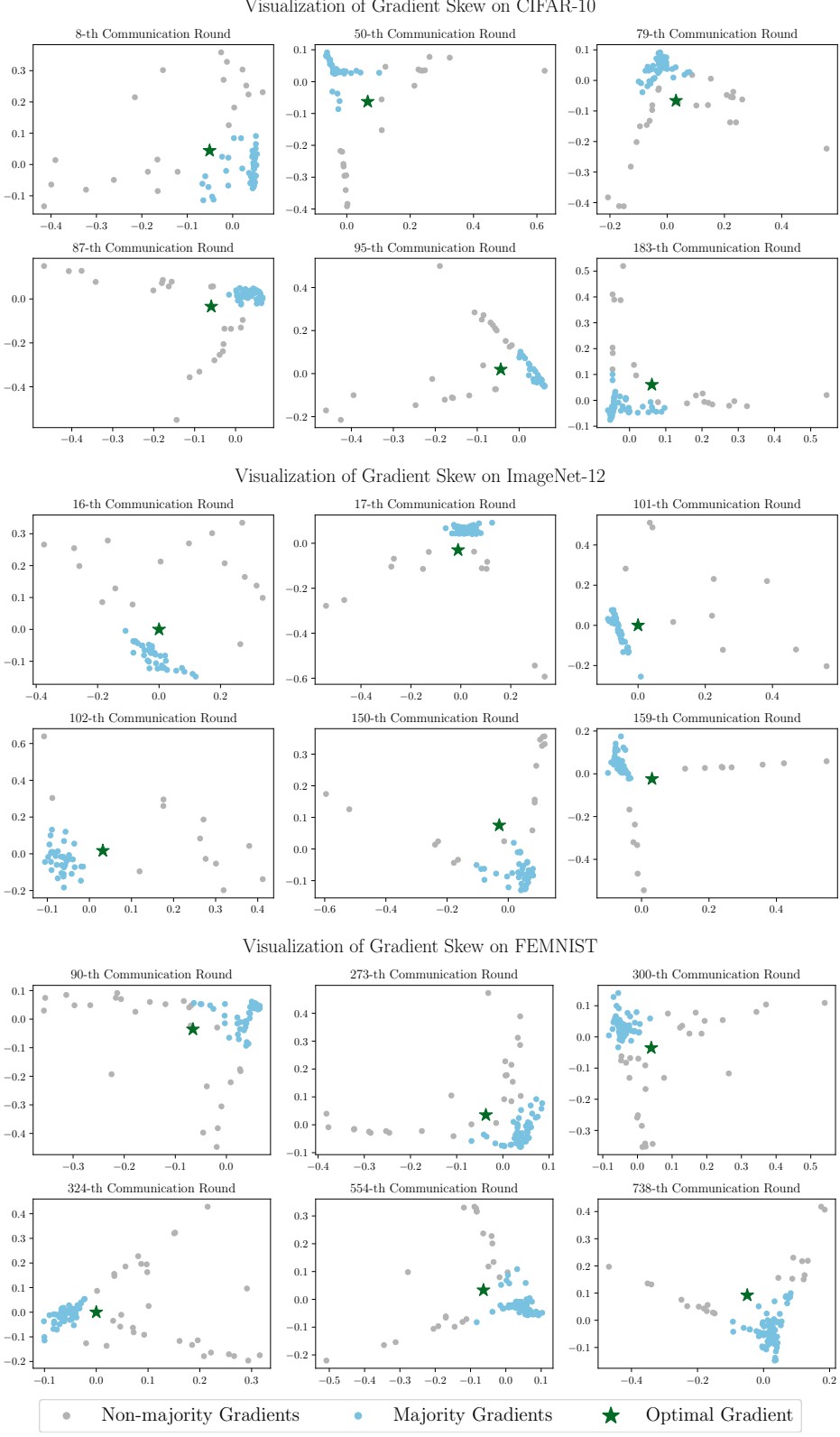

Figure 4: Visualization of gradient skew on three benchmark datasets.

Visualization of STRIKE Attack on CIFAR-10

Figure 5: Visualization of STRIKE attack on CIFAR-10 datasets. The visualization shows that Byzantine gradients can hide within the skewed honest gradients well, which justifies that the heuristic search in the first stage of STRIKE attack can effectively find the skewed honest gradients.

**Definition 2** ($(f, \gamma)$-skewed). The set of honest gradients $\{g_i \mid i \in \mathcal{H}\}$ is called $(f, \gamma)$-skewed if there exists a set $\mathcal{S} \subseteq \mathcal{H}$ of size $n - 2f$ such that

$$\mathbb{E}[\|\bar{g}_{\mathcal{S}} - \bar{g}\|^2] \geq \gamma \rho_{\mathcal{S}}^2, \tag{21}$$

where $\bar{g} = \sum_{i \in \mathcal{H}} g_i/(n-f)$, $\bar{g}_{\mathcal{S}} = \sum_{i \in \mathcal{S}} g_i/(n-2f)$, and $\rho_{\mathcal{S}}^2 = \mathbb{E}[\max_{i,j \in \mathcal{S}}\|g_i - g_j\|^2]$ is a measure of gradient heterogeneity introduced by El-Mhamdi et al. (2021). Here, gradients $\{g_i \mid i \in \mathcal{S}\}$ are called the *skewed honest gradients* (of honest gradients), and $\gamma$ is called the skewness of honest gradients $\{g_i \mid i \in \mathcal{H}\}$.

Below, we provide an example to help have a better understanding of gradient skew.

**Example.** We consider the following simple example where there are $n$ clients in total of which $f$ are Byzantine and $n - f$ are honest. Assume that

- for client $i \in \{1, ..., f\}$ holding gradient $g_i \sim \mathcal{N}(\mu_1, \sigma^2 I)$.
- for client $i \in \{f + 1, ..., n - 2f\}$ holding gradient $g_i \sim \mathcal{N}(\mu_2, \sigma^2 I)$.

Here, $\sim$ represents a random vector following a distribution, $\mathcal{N}(\mu, \sigma^2 I)$ represents the normal distribution with mean $\mu$ and covariance matrix $\sigma^2 I$, $\mu_1$ and $\mu_2$ are vectors, $\sigma^2 > 0$ is the variance and $I$ represents identity matrix.

In this example, the skewness is

$$\gamma = \frac{1}{2}\left(\frac{1}{n - f} + \frac{1}{n - 2f} + \frac{f^2}{(n - f)^2} \cdot \frac{\|\mu_1 - \mu_2\|^2}{\sigma^2}\right). \tag{22}$$

**Assumption 1** ($L$-smooth). The loss function is $L$-smooth, i.e.,

$$\|\nabla \mathcal{L}(w) - \nabla \mathcal{L}(w')\| \leq \|w - w'\|, \quad \forall w, w' \in \mathbb{R}^d. \tag{23}$$

**Assumption 2** (Unbias). The stochastic gradients sampled from any local data distribution are unbiased estimators of local gradients for all clients, i.e.,

$$\mathbb{E}[g_i^t] = \nabla \mathcal{L}(w^t), \quad \forall i = 1, \ldots n, t = 0, \ldots, T - 1. \tag{24}$$

## B.1 PROOFS

### B.1.1 SUPPORTING LEMMA

We start with proving the lemma stated below.

**Lemma 1.** *Given any $d$-dimensional random vectors $\boldsymbol{X}$ and $\boldsymbol{Y}$, the following inequalities hold:*

$$(\sqrt{\mathbb{E}[\|\boldsymbol{X}\|^2]} - \sqrt{\mathbb{E}[\|\boldsymbol{Y}\|^2]})^2 \le \mathbb{E}[\|\boldsymbol{X} + \boldsymbol{Y}\|^2] \le (\sqrt{\mathbb{E}[\|\boldsymbol{X}\|^2]} + \sqrt{\mathbb{E}[\|\boldsymbol{Y}\|^2]})^2 \tag{25}$$

*Proof.* $\mathbb{E}[\|\boldsymbol{X} + \boldsymbol{Y}\|^2]$ can be written as follows,

$$\mathbb{E}[\|\boldsymbol{X} + \boldsymbol{Y}\|^2] = \mathbb{E}[\|\boldsymbol{X}\|^2 + \|\boldsymbol{Y}\|^2 + 2\langle \boldsymbol{X}, \boldsymbol{Y} \rangle] = \mathbb{E}[\|\boldsymbol{X}\|^2] + \mathbb{E}[\|\boldsymbol{Y}\|^2] + 2\mathbb{E}[\langle \boldsymbol{X}, \boldsymbol{Y} \rangle]. \tag{26}$$

According to the Cauchy–Schwarz inequality, we have

$$|\mathbb{E}[\langle \boldsymbol{X}, \boldsymbol{Y} \rangle]| \le \mathbb{E}[|\langle \boldsymbol{X}, \boldsymbol{Y} \rangle|] \le \mathbb{E}[\|\boldsymbol{X}\|\|\boldsymbol{Y}\|] \le \sqrt{\mathbb{E}[\|\boldsymbol{X}\|^2]\mathbb{E}[\|\boldsymbol{Y}\|^2]}. \tag{27}$$

That is

$$-\sqrt{\mathbb{E}[\|\boldsymbol{X}\|^2]\mathbb{E}[\|\boldsymbol{Y}\|^2]} \le \mathbb{E}[\langle \boldsymbol{X}, \boldsymbol{Y} \rangle] \le \sqrt{\mathbb{E}[\|\boldsymbol{X}\|^2]\mathbb{E}[\|\boldsymbol{Y}\|^2]}. \tag{28}$$

Combine Equation (26) and Inequality (28), then we have

$$\mathbb{E}[\|\boldsymbol{X} + \boldsymbol{Y}\|^2] \ge \mathbb{E}[\|\boldsymbol{X}\|^2] + \mathbb{E}[\|\boldsymbol{Y}\|^2] - 2\sqrt{\mathbb{E}[\|\boldsymbol{X}\|^2]\mathbb{E}[\|\boldsymbol{Y}\|^2]} = (\sqrt{\mathbb{E}[\|\boldsymbol{X}\|^2]} - \sqrt{\mathbb{E}[\|\boldsymbol{Y}\|^2]})^2, \tag{29}$$

and

$$\mathbb{E}[\|\boldsymbol{X} + \boldsymbol{Y}\|^2] \le \mathbb{E}[\|\boldsymbol{X}\|^2] + \mathbb{E}[\|\boldsymbol{Y}\|^2] + 2\sqrt{\mathbb{E}[\|\boldsymbol{X}\|^2]\mathbb{E}[\|\boldsymbol{Y}\|^2]} = (\sqrt{\mathbb{E}[\|\boldsymbol{X}\|^2]} + \sqrt{\mathbb{E}[\|\boldsymbol{Y}\|^2]})^2. \tag{30}$$

$\square$

### B.1.2 PROOF OF PROPOSITION 1

We recall the proposition statement below.

**Proposition 1** (Vulnerability under skew). *Given any $(f, \lambda)$-resilient AGR $\mathcal{A}$, $\gamma = \Omega(\lambda^2)$, if the set of honest gradients $\{\boldsymbol{g}_i \mid i \in \mathcal{H}\}$ is $(f, \gamma)$-skewed, then there exist Byzantine gradients $\{\boldsymbol{g}_i \mid i \in \mathcal{B}\}$ such that*

$$\mathbb{E}[\|\mathcal{A}(\boldsymbol{g}_1, \ldots, \boldsymbol{g}_n) - \bar{\boldsymbol{g}}\|^2] \ge \Omega(\frac{\gamma}{\lambda^2} \cdot \frac{f^2}{(n-f)^2} \cdot \rho_{\mathcal{S}}^2). \tag{31}$$

*where $\bar{\boldsymbol{g}} = \sum_{i \in \mathcal{H}} \boldsymbol{g}_i / (n-f)$ is the optimal gradient, $\rho_{\mathcal{S}}^2 = \mathbb{E}[\max_{i,j \in \mathcal{S}} \|\boldsymbol{g}_i - \boldsymbol{g}_j\|^2]$, $\mathcal{S}$ is the index set of the skewed honest gradients.*

*Proof.* According to Definition 2, there exists $\mathcal{S} \subseteq \mathcal{H}$ of size $n - 2f$ and $\gamma > 1$ such that

$$\mathbb{E}[\|\bar{\boldsymbol{g}}_{\mathcal{S}} - \bar{\boldsymbol{g}}\|^2] = \gamma \rho_{\mathcal{S}}^2. \tag{32}$$

For all $i \in \mathcal{B}$, we set Byzantine gradient $\boldsymbol{g}_i = \bar{\boldsymbol{g}}_{\mathcal{S}}$. We then show that, under this attack, the aggregation error is lower-bounded as shown in Equation (8).

We consider the average and heterogeneity of the forged honest gradients $\{\boldsymbol{g}_i \mid i \in \mathcal{S} \cup \mathcal{B}\}$.

The average is computed as follows.

$$\bar{\boldsymbol{g}}_{\mathcal{B} \cup \mathcal{S}} = \frac{1}{n-f} \sum_{i \in \mathcal{B} \cup \mathcal{S}} \boldsymbol{g}_i \tag{33}$$

$$= \frac{1}{n-f} (\sum_{i \in \mathcal{B}} \boldsymbol{g}_i + \sum_{i \in \mathcal{S}} \boldsymbol{g}_i) \tag{34}$$

$$= \frac{1}{n-f} (f \bar{\boldsymbol{g}}_{\mathcal{S}} + (n - 2f) \bar{\boldsymbol{g}}_{\mathcal{S}}) \tag{35}$$

$$= \bar{\boldsymbol{g}}_{\mathcal{S}}. \tag{36}$$

Then we consider the heterogeneity of gradients $\{\boldsymbol{g}_i \mid i \in \mathcal{S} \cup \mathcal{B}\}$ $\rho_{\mathcal{S} \cup \mathcal{B}}$.

For all $b \in \mathcal{B}$ and $i \in \mathcal{S}$,

$$\|\boldsymbol{g}_b - \boldsymbol{g}_i\|^2 = \|\bar{\boldsymbol{g}}_{\mathcal{S}} - \boldsymbol{g}_i\|^2 \tag{37}$$

$$= \|\frac{1}{n-2f} \sum_{j \in \mathcal{S}} \boldsymbol{g}_j - \boldsymbol{g}_i\|^2 \tag{38}$$

$$= \|\frac{1}{n-2f} \sum_{j \in \mathcal{S}} (\boldsymbol{g}_j - \boldsymbol{g}_i)\|^2 \tag{39}$$

$$\leq \frac{1}{n-2f} \sum_{j \in \mathcal{S}} \|\boldsymbol{g}_j - \boldsymbol{g}_i\|^2 \tag{40}$$

$$\leq \max_{j \in \mathcal{S}} \|\boldsymbol{g}_j - \boldsymbol{g}_i\|^2 \tag{41}$$

where Inequality (40) comes from the Cauchy inequality.

Then for the heterogeneity of $\{\boldsymbol{g}_i \mid i \in \mathcal{S} \cup \mathcal{B}\}$, we have:

$$\rho_{\mathcal{B} \cup \mathcal{S}}^2 = \mathbb{E}[\max_{i,j \in \mathcal{B} \cup \mathcal{S}} \|\boldsymbol{g}_i - \boldsymbol{g}_j\|^2] \tag{42}$$

$$= \mathbb{E}[\max_{i,j \in \mathcal{S}} \|\boldsymbol{g}_i - \boldsymbol{g}_j\|^2] \tag{43}$$

$$= \rho_{\mathcal{S}}^2. \tag{44}$$

For notation simplicity, we denote $\mathcal{A}(\boldsymbol{g}_1, \ldots, \boldsymbol{g}_n)$ by $\hat{\boldsymbol{g}}$. Then we can lower bound $\mathbb{E}[\|\hat{\boldsymbol{g}} - \bar{\boldsymbol{g}}\|^2]$ as follows

$$\mathbb{E}[\|\hat{\boldsymbol{g}} - \bar{\boldsymbol{g}}\|^2] = \mathbb{E}[\|(\bar{\boldsymbol{g}} - \bar{\boldsymbol{g}}_{\mathcal{S} \cup \mathcal{B}}) - (\hat{\boldsymbol{g}} - \bar{\boldsymbol{g}}_{\mathcal{S} \cup \mathcal{B}}\|))^2] \tag{45}$$

$$= \mathbb{E}[\|(\bar{\boldsymbol{g}} - \bar{\boldsymbol{g}}_{\mathcal{S}}) - (\hat{\boldsymbol{g}} - \bar{\boldsymbol{g}}_{\mathcal{S} \cup \mathcal{B}}\|))^2] \tag{46}$$

$$\geq (\sqrt{\mathbb{E}[\|\bar{\boldsymbol{g}} - \bar{\boldsymbol{g}}_{\mathcal{S}}\|]^2} - \sqrt{\mathbb{E}[\|\hat{\boldsymbol{g}} - \bar{\boldsymbol{g}}_{\mathcal{S} \cup \mathcal{B}}\|^2]})^2. \tag{47}$$

Here, Equation (46) is due to Equation (33), Inequality (47) relies on Lemma 1

We can lower bound term $\sqrt{\mathbb{E}[\|\bar{\boldsymbol{g}} - \bar{\boldsymbol{g}}_{\mathcal{S}}\|^2]} - \sqrt{\mathbb{E}[\|\hat{\boldsymbol{g}} - \bar{\boldsymbol{g}}_{\mathcal{S} \cup \mathcal{B}}\|^2]}$ as follows.

$$\sqrt{\mathbb{E}[\|\bar{\boldsymbol{g}} - \bar{\boldsymbol{g}}_{\mathcal{S}}\|^2]} - \sqrt{\mathbb{E}[\|\hat{\boldsymbol{g}} - \bar{\boldsymbol{g}}_{\mathcal{S} \cup \mathcal{B}}\|^2]} \geq \sqrt{\gamma \cdot \rho_{\mathcal{S}}^2} - \sqrt{\lambda^2 \rho_{\mathcal{S} \cup \mathcal{B}}^2} \tag{48}$$

$$= \sqrt{\gamma \cdot \rho_{\mathcal{S}}^2} - \sqrt{\lambda^2 \rho_{\mathcal{S}}^2} \tag{49}$$

$$= (\frac{\sqrt{\gamma}}{\lambda} - 1)\lambda \rho_{\mathcal{S}} \tag{50}$$

$$\geq (\frac{\sqrt{\gamma}}{\lambda} - 1)\frac{f}{n-f} \cdot \rho_{\mathcal{S}} \tag{51}$$

$$= \Omega(\frac{\sqrt{\gamma}}{\lambda} \cdot \frac{f}{n-f} \cdot \rho_{\mathcal{S}}) \tag{52}$$

where Equation (48) results from Equation (32) and Equation (6), Equation (49) relies on Equation (44). In Inequality (51), we use the fact $\lambda \geq f/(n-f)$ from Farhadkhani et al. (2022).

We combine Inequality (47) and Equation (52) for the final conclusion in Equation (8):

$$\mathbb{E}[\|\hat{\boldsymbol{g}} - \bar{\boldsymbol{g}}\|^2] = \Omega(\frac{\gamma}{\lambda^2} \cdot \frac{f^2}{(n-f)^2} \cdot \rho_{\mathcal{S}}^2). \tag{53}$$

$\square$

### B.1.3 PROOF FOR PROPOSITION 2

We recall the proposition statement below.

**Proposition 2.** *Given any $(f,\lambda)$-resilient AGR $\mathcal{A}$, $L$-smooth global loss function $\mathcal{L}$, and learning rate $\eta \leq 1/L$, $\gamma = \Omega(\lambda^2)$, if honest gradients $\{\boldsymbol{g}_i^t \mid i \in \mathcal{H}\}$ are $(f,\gamma)$-skewed for all $t = 0,\ldots,T-1$, then there exists Byzantine gradients $\{\boldsymbol{g}_b^t \mid b \in \mathcal{B}, t = 0,\ldots,T-1\}$ such that the global model parameter is bounded away from the global optimum $\boldsymbol{w}^*$:*

$$\mathbb{E}[\|\boldsymbol{w}^t - \boldsymbol{w}^*\|^2] \geq \Omega(\eta^2(1 - L\eta)^2 \cdot \frac{\gamma}{\lambda^2} \cdot \frac{f^2}{(n-f)^2} \cdot \rho^2), \quad t = 1,\ldots,T, \tag{54}$$

*where $\boldsymbol{w}^t$ is the parameter of global model in the $t$-th communication round, $\boldsymbol{w}^*$ is the global optimum of global loss function $\mathcal{L}$, $\rho^2 = \min_{t=0,\ldots,T-1} \mathbb{E}[\max_{i,j \in \mathcal{S}^t} \|\boldsymbol{g}_i^t - \boldsymbol{g}_j^t\|^2]$, and $\mathcal{S}^t$ is the index set of the skewed honest gradients in $t$-th communication round.*

*Remark* 1. In practice, the learning rate is typically vanishing with increasing the training round $T$ to ensure convergence (Karimireddy et al., 2022). Therefore, the lower bound in Equation (54) vanishes with the increasing communication rounds, which is *consistent* with the upper bound in Theorem V in (Karimireddy et al., 2022) that shows the model will finally converge to a stationary point under arbitrary attack when applying a vanishing learning rate. In fact, no Byzantine attack can break the upper bound in (Karimireddy et al., 2022). The lower bound, although vanishes with increasing communication rounds, can still show how much the attack can *hinder* the training process. Thus, it is still meaningful in the field of Byzantine robustness.

*Proof.* According to Proposition 1, for all $t = 0,\ldots,T-1$, there exist Byzantine gradients $\{\boldsymbol{g}_i^t \mid i \in \mathcal{B}\}$ such that

$$\mathbb{E}[\|\hat{\boldsymbol{g}}^t - \bar{\boldsymbol{g}}^t\|^2] \geq C \cdot \frac{\gamma}{\lambda^2} \cdot \frac{f^2}{(n-f)^2} \cdot (\rho^t)^2, \tag{55}$$

where $C$ is a constant, and $(\rho^t)^2 = \max_{i,j \in \mathcal{S}^t} \|\boldsymbol{g}_i^t - \boldsymbol{g}_j^t\|^2$, and $\mathcal{S}^t$ is the skewed honest gradients in $t$-th communication round. Let $\rho^2 = \min_{t=1,\ldots,T-1}(\rho^t)^2$, then we have

$$\mathbb{E}[\|\hat{\boldsymbol{g}}^t - \bar{\boldsymbol{g}}^t\|^2] \geq C \cdot \frac{\gamma}{\lambda^2} \cdot \frac{f^2}{(n-f)^2} \cdot \rho^2, \tag{56}$$

We prove Equation (8) in the following two different cases.

**Case 1.** $\mathbb{E}[\|\boldsymbol{w}^t - \boldsymbol{w}^*\|^2] < C\eta^2\gamma f^2\rho^2/4\lambda^2(n-f)^2$.

Since $\boldsymbol{w}^{t+1} = \boldsymbol{w}^t - \eta\hat{\boldsymbol{g}}^t$, we can rewrite $\|\boldsymbol{w}^{t+1} - \boldsymbol{w}^*\|^2$ as follows.

$$\|\boldsymbol{w}^{t+1} - \boldsymbol{w}^*\|^2 = \|(\boldsymbol{w}^t - \eta\hat{\boldsymbol{g}}^t) - \boldsymbol{w}^*\|^2 \tag{57}$$

$$= \|(\nabla\mathcal{L}(\boldsymbol{w}^t) - \eta\hat{\boldsymbol{g}}^t) + (\boldsymbol{w}^t - \boldsymbol{w}^* - \eta\nabla\mathcal{L}(\boldsymbol{w}^t))\|^2 \tag{58}$$

$$= \|(\nabla\mathcal{L}(\boldsymbol{w}^t) - \eta\hat{\boldsymbol{g}}^t) + (\boldsymbol{w}^t - \boldsymbol{w}^* - \eta(\nabla\mathcal{L}(\boldsymbol{w}^t) - \nabla\mathcal{L}(\boldsymbol{w}^*)))\|^2. \tag{59}$$

In Equation (59) we use the fact that $\nabla\mathcal{L}(\boldsymbol{w}^*) = \boldsymbol{0}$.

Combine Equation (59) and Lemma 1, we can lower bound $\mathbb{E}[\|\boldsymbol{w}^{t+1} - \boldsymbol{w}^*\|^2]$ as follows,

$$\mathbb{E}[\|\boldsymbol{w}^{t+1} - \boldsymbol{w}^*\|^2] = \|(\nabla\mathcal{L}(\boldsymbol{w}^t) - \eta\hat{\boldsymbol{g}}^t) + (\boldsymbol{w}^t - \boldsymbol{w}^* - \eta(\nabla\mathcal{L}(\boldsymbol{w}^t) - \nabla\mathcal{L}(\boldsymbol{w}^*)))\|^2 \tag{60}$$

$$\geq (\eta\underbrace{\sqrt{\mathbb{E}[\|\nabla\mathcal{L}(\boldsymbol{w}^t) - \hat{\boldsymbol{g}}^t\|^2]}}_{A} - \underbrace{\sqrt{\mathbb{E}[\|\boldsymbol{w}^t - \boldsymbol{w}^* - \eta(\nabla\mathcal{L}(\boldsymbol{w}^t) - \nabla\mathcal{L}(\boldsymbol{w}^*))\|^2]}}_{B})^2. \tag{61}$$

To obtain a further lower bound for Equation (61) amounts to give lower and upper bound for terms $A$ and $B$, respectively.

To lower bound term $A$, again we use Lemma 1,

$$\mathbb{E}[\|\nabla\mathcal{L}(\boldsymbol{w}^t) - \hat{\boldsymbol{g}}^t\|^2] = \mathbb{E}[\|(\bar{\boldsymbol{g}}^t - \hat{\boldsymbol{g}}^t) + (\nabla\mathcal{L}(\boldsymbol{w}^t) - \bar{\boldsymbol{g}}^t)\|^2] \tag{62}$$

$$\geq (\sqrt{\mathbb{E}[\|\bar{\boldsymbol{g}}^t - \hat{\boldsymbol{g}}^t\|^2]} - \sqrt{\mathbb{E}[\|\nabla\mathcal{L}(\boldsymbol{w}^t) - \bar{\boldsymbol{g}}^t\|^2]})^2 \tag{63}$$

$$\geq (\sqrt{C} \cdot \frac{\sqrt{\gamma}}{\lambda} \cdot \frac{f}{n-f} \cdot \rho - \frac{\sigma}{\sqrt{n-f}})^2. \tag{64}$$

Here, $\sigma^2 = \sum_{i \in \mathcal{H}} \mathrm{Var}[\boldsymbol{g}_i^t]/(n-f)$ is the average variance of stochastic gradients. Inequality (64) is a combined result of Equation (56) and the law of large numbers.

We apply Lemma 1 to upper-bound term B as follows,

$$\mathbb{E}[\|\boldsymbol{w}^t - \boldsymbol{w}^* - (\nabla\mathcal{L}(\boldsymbol{w}^t) - \nabla\mathcal{L}(\boldsymbol{w}^*))\|^2] \le \mathbb{E}[(\|\boldsymbol{w}^t - \boldsymbol{w}^*\| + \eta \cdot L\|\boldsymbol{w}^t - \boldsymbol{w}^*\|)^2] \tag{65}$$

$$= (1 + L\eta)^2 \mathbb{E}[\|\boldsymbol{w}^t - \boldsymbol{w}^*\|^2] \tag{66}$$

$$\le (1 + L\eta)^2 \cdot \frac{C}{4} \cdot \eta^2 \cdot \frac{\gamma}{\lambda^2} \cdot \frac{f^2}{(n-f)^2} \cdot \rho^2 \tag{67}$$

Combine Inequality (64) and Inequality (67), we have

$$\mathbb{E}[\|\boldsymbol{w}^{t+1} - \boldsymbol{w}^*\|^2] \ge (\eta(\sqrt{C} \cdot \frac{\sqrt{\gamma}}{\lambda} \cdot \frac{f}{n-f} \cdot \rho - \frac{\sigma}{\sqrt{n-f}}) - \frac{\eta(1+L\eta)}{2} \cdot \frac{\sqrt{C\gamma}}{\lambda} \cdot \frac{f}{n-f} \cdot \rho)^2 \tag{68}$$

$$= (\frac{\eta(1-L\eta)}{2} \cdot \frac{\sqrt{C\gamma}}{\lambda} \cdot \frac{f}{n-f} \cdot \rho - \frac{\sigma}{\sqrt{n-f}})^2 \tag{69}$$

$$= \Omega(\eta^2(1-L\eta)^2 \cdot \frac{\gamma}{\lambda^2} \cdot \frac{f^2}{(n-f)^2} \cdot \rho^2) \tag{70}$$

Here Equation (70) uses the fact that SGD variance $\sigma^2$ is negligible with respect to the gradient heterogeneity $\rho^2$.

**Case 2.** $\mathbb{E}[\|\boldsymbol{w}^t - \boldsymbol{w}^*\|^2] \ge C\eta^2\gamma f^2\rho^2/4\lambda^2(n-f)^2$. In this case, we let Byzantine gradients behave honestly such that $\hat{\boldsymbol{g}}^t = \bar{\boldsymbol{g}}^t$. Then $\mathbb{E}[\|\boldsymbol{w}^{t+1} - \boldsymbol{w}^*\|^2]$ can be lower-bounded as follows.

$$\mathbb{E}[\|\boldsymbol{w}^{t+1} - \boldsymbol{w}^*\|^2] = \mathbb{E}[\|(\boldsymbol{w}^t - \eta\bar{\boldsymbol{g}}^t) - \boldsymbol{w}^*\|^2] \tag{71}$$

$$= \mathbb{E}[\|\boldsymbol{w}^t - \boldsymbol{w}^* - \eta(\nabla\mathcal{L}(\boldsymbol{w}^t) - \nabla\mathcal{L}(\boldsymbol{w}^*)) - \eta(\bar{\boldsymbol{g}}^t - \nabla\mathcal{L}(\boldsymbol{w}^t))\|^2] \tag{72}$$

$$\ge (\sqrt{E[\|\boldsymbol{w}^t - \boldsymbol{w}^* - \eta(\nabla\mathcal{L}(\boldsymbol{w}^t) - \nabla\mathcal{L}(\boldsymbol{w}^*))\|^2]} - \eta\sqrt{\mathbb{E}[\|\bar{\boldsymbol{g}}^t - \nabla\mathcal{L}(\boldsymbol{w}^t)\|^2]})^2. \tag{73}$$

In Equation (72) we use the fact that $\nabla\mathcal{L}(\boldsymbol{w}^*) = \boldsymbol{0}$, and Equation (73) comes from Lemma 1

We first lower-bound $\mathbb{E}[\|\boldsymbol{w}^t - \boldsymbol{w}^* - \eta(\nabla\mathcal{L}(\boldsymbol{w}^t) - \nabla\mathcal{L}(\boldsymbol{w}^*))\|^2]$,

$$\mathbb{E}[\|\boldsymbol{w}^t - \boldsymbol{w}^* - \eta(\nabla\mathcal{L}(\boldsymbol{w}^t) - \nabla\mathcal{L}(\boldsymbol{w}^*))\|^2] \ge \mathbb{E}[(\|\boldsymbol{w}^t - \boldsymbol{w}^*\| - \eta \cdot L\|\boldsymbol{w}^t - \boldsymbol{w}^*\|)^2] \tag{74}$$

$$= (1 - L\eta)^2 \mathbb{E}[\|\boldsymbol{w}^t - \boldsymbol{w}^*\|^2] \tag{75}$$

$$\ge (1 - L\eta)^2 \cdot \frac{C}{4} \cdot \eta^2 \cdot \frac{\gamma}{\lambda^2} \cdot \frac{f^2}{(n-f)^2} \cdot \rho^2 \tag{76}$$

Then we upper-bound $\mathbb{E}[\|\hat{\boldsymbol{g}}^t - \nabla\mathcal{L}(\boldsymbol{w}^t)\|^2]$

$$\mathbb{E}[\|\hat{\boldsymbol{g}}^t - \nabla\mathcal{L}(\boldsymbol{w}^t)\|^2] = \mathbb{E}[\|\bar{\boldsymbol{g}}^t - \nabla\mathcal{L}(\boldsymbol{w}^t)\|^2] \tag{77}$$

$$\le \frac{\sigma^2}{n-f} \tag{78}$$

Here, $\sigma^2 = \sum_{i \in \mathcal{H}} \mathrm{Var}[\boldsymbol{g}_i^t]/(n-f)$ is the average variance of stochastic gradients.

Combining Equation (76) and Equation (78), we have

$$\mathbb{E}[\|\boldsymbol{w}^{t+1} - \boldsymbol{w}^*\|^2] \ge (\frac{\eta(1-L\eta)}{2} \cdot \frac{\sqrt{C\gamma}}{\lambda} \cdot \frac{f}{n-f} \cdot \rho - \eta\frac{\sigma}{\sqrt{n-f}})^2 \tag{79}$$

$$= \Omega(\eta^2(1-L\eta)^2 \cdot \frac{\gamma}{\lambda^2} \cdot \frac{f^2}{(n-f)^2} \cdot \rho^2) \tag{80}$$

Here Equation (80) uses the fact that SGD variance $\sigma^2$ is negligible with respect to the gradient heterogeneity $\rho^2$.

In both cases, we have

$$\mathbb{E}[\|\boldsymbol{w}^{t+1} - \boldsymbol{w}^*\|^2] = \Omega(\eta^2(1 - L\eta)^2 \cdot \frac{\gamma}{\lambda^2} \cdot \frac{f^2}{(n-f)^2} \cdot \rho^2), \quad t = 0, \dots, T-1, \qquad (81)$$

which completes the proof.

$\square$

## B.2 APPLICATION TO OTHER DEFINITIONS OF BYZANTINE RESILIENCE

In this section, we discuss how our analysis applies to other definitions of Byzantine resilience. In particular, we consider the definitions of Byzantine resilience in recent works of Karimireddy et al. (2022); Allouah et al. (2023).

### B.2.1 CIRCUMVENT $(\delta_{\max}, c)$-AGRS

The following formulation of Byzantine resilience in Karimireddy et al. (2022) improves the upper bound by the fraction of Byzantine clients, and thus can recover the standard convergence rate when there are no Byzantine clients.

**Definition 3** ($(\delta_{\max}, c)$-AGR). *A robust AGR $gA$ is called a $(\delta_{\max}, c)$-AGR if, given any input $\{\boldsymbol{g}_1, \dots, \boldsymbol{g}_n\}$ of which a subset of at least size $|\mathcal{G}| > (1 - \delta)n$ for $\delta \leq \delta_{\max} < 0.5$ and satisfies $\mathbb{E}[\|\boldsymbol{g}_i - \boldsymbol{g}_j\|] \leq \rho^2$, the output $\hat{\boldsymbol{g}}$ of AGR $\mathcal{A}$ satisfies:*

$$\mathbb{E}[\|\hat{\boldsymbol{g}} - \bar{\boldsymbol{g}}_{\mathcal{G}}\|^2] \leq c\delta\rho^2 \quad \text{where} \quad \hat{\boldsymbol{g}} = \mathcal{A}_\delta(\boldsymbol{g}_1, \dots, \boldsymbol{g}_n), \bar{\boldsymbol{g}}_{\mathcal{G}} = \sum_{i \in \mathcal{G}} \boldsymbol{g}_i/(n-f). \qquad (82)$$

We show that any $(\delta_{\max}, c)$-AGR $\mathcal{A}$ also satisfies the resilience defined in Definition 1

**Proposition 3.** *Any $(\delta_{\max}, c)$-AGR $\mathcal{A}$ is $(f, \lambda)$-resilient for any $f \leq \delta_{\max}n$ and $\lambda = \sqrt{c\delta}$.*

*Proof.* Consider any deterministic vectors $\{\boldsymbol{g}_1, \dots, \boldsymbol{g}_n\}$, $f \leq \delta_{\max}n$, and subset $\mathcal{G} \subseteq \{1, \dots, n\}$ of size $n - f$. According to Definition 3, we have

$$\|\hat{\boldsymbol{g}} - \bar{\boldsymbol{g}}_{\mathcal{G}}\|^2 \leq c\delta\rho^2 \qquad (83)$$

where $\hat{\boldsymbol{g}} = \mathcal{A}_\delta(\boldsymbol{g}_1, \dots, \boldsymbol{g}_n)$, $\bar{\boldsymbol{g}}_{\mathcal{G}} = \sum_{i \in \mathcal{G}} \boldsymbol{g}_i/(n-f)$, $\delta = f/n$, and $\rho^2 \geq \max_{i,j \in \mathcal{G}} \|\boldsymbol{g}_i - \boldsymbol{g}_j\|^2$ The expectation is dropped since input vectors $\{\boldsymbol{g}_1, \dots, \boldsymbol{g}_n\}$, $f \leq \delta_{\max}n$ are deterministic. We take $\rho^2 = \max_{i,j \in \mathcal{G}} \|\boldsymbol{g}_i - \boldsymbol{g}_j\|$ take the square root of both sides of Inequality (83), then we have

$$\|\hat{\boldsymbol{g}} - \bar{\boldsymbol{g}}_{\mathcal{G}}\| \leq \sqrt{c\delta} \max_{i,j \in \mathcal{G}} \|\boldsymbol{g}_i - \boldsymbol{g}_j\|. \qquad (84)$$

Therefore, $\mathcal{A}$ is $(f, \lambda)$-resilient for any $f \leq \delta_{\max}n$ and $\lambda = \sqrt{cf/n}$. $\square$

Combining Proposition 3 with Proposition 1 and Proposition 2, the following corollaries are obvious.

**Corollary 1.** *Given any $(\delta_{\max}, c)$-AGR $\mathcal{A}$ with $\delta_{\max} \geq f/n$, if the set of honest gradients $\{\boldsymbol{g}_i \mid i \in \mathcal{H}\}$ is $(f, \gamma)$-skewed, then there exist Byzantine gradients $\{\boldsymbol{g}_i \mid i \in \mathcal{B}\}$ such that*

$$\mathbb{E}[\|\mathcal{A}(\boldsymbol{g}_1, \dots, \boldsymbol{g}_n) - \bar{\boldsymbol{g}}\|^2] \geq \Omega(\frac{\gamma}{c} \cdot \frac{f}{n-f} \cdot \rho_{\mathcal{S}}^2). \qquad (85)$$

*where $\bar{\boldsymbol{g}} = \sum_{i \in \mathcal{H}} \boldsymbol{g}_i/(n-f)$ is the optimal gradient, $\rho_{\mathcal{S}}^2 = \mathbb{E}[\max_{i,j \in \mathcal{S}} \|\boldsymbol{g}_i - \boldsymbol{g}_j\|^2]$, $\mathcal{S}$ is the index set of the skewed honest gradients.*

**Corollary 2.** *Given any $(\delta_{\max}, c)$-resilient AGR $\mathcal{A}$ with $\delta_{\max} \geq f/n$, L-smooth global loss function $\mathcal{L}$, and learning rate $\eta \leq 1/L$, if honest gradients $\{\boldsymbol{g}_i^t \mid i \in \mathcal{H}\}$ are $(f, \gamma)$-skewed for all $t = 0, \dots, T-1$, then there exists Byzantine gradients $\{\boldsymbol{g}_b^t \mid b \in \mathcal{B}, t = 0, \dots, T-1\}$ such that the global model parameter is bounded away from the global optimum $\boldsymbol{w}^*$:*

$$\mathbb{E}[\|\boldsymbol{w}^t - \boldsymbol{w}^*\|^2] \geq \Omega(\eta^2(1 - L\eta)^2 \cdot \frac{\gamma}{c} \cdot \frac{f}{n-f} \cdot \rho^2), \quad t = 1, \dots, T, \qquad (86)$$

*where $\boldsymbol{w}^t$ is the parameter of global model in the $t$-th communication round, and $\boldsymbol{w}^*$ is the global optimum of global loss function $\mathcal{L}$.*

### B.2.2 Circumvent $(f, \kappa)$-robust AGRs

The following notion of Byzantine resilience in Allouah et al. (2023) is also a unified robustness criterion that is fine-grained to obtain tight convergence guarantees.

**Definition 4** $((f, \kappa)$-robust). *Let $f < n/2$ and $\kappa \geq 0$, a robust AGR $gA$ is called $(f, \kappa)$-robust] if for any input $\{\boldsymbol{g}_1, \ldots, \boldsymbol{g}_n\}$ and any set $\mathcal{G} \subseteq \mathcal{G}$ of size $n - f$, the output $\hat{\boldsymbol{g}}$ of AGR $\mathcal{A}$ satisfies:*

$$\|\mathcal{A}(\boldsymbol{g}_1, \ldots, \boldsymbol{g}_n) - \bar{\boldsymbol{g}}_{\mathcal{G}}\| \leq \frac{\kappa}{n - f} \sum_{i \in \mathcal{S}} \|\boldsymbol{g}_i - \bar{\boldsymbol{g}}_{\mathcal{G}}\|^2 \quad \text{where} \quad \bar{\boldsymbol{g}}_{\mathcal{G}} = \sum_{i \in \mathcal{G}} \boldsymbol{g}_i / (n - f). \quad (87)$$

We show that any $(f, \kappa)$-robust $\mathcal{A}$ also satisfies the resilience defined in Definition 1.

**Proposition 4.** *Any $(f, \kappa)$-robust AGR $\mathcal{A}$ is $(f, \lambda)$-resilient for $\lambda = \sqrt{\kappa}$.*

*Proof.* Given any deterministic vectors $\{\boldsymbol{g}_1, \ldots, \boldsymbol{g}_n\}$ and subset $\mathcal{G} \subseteq \{1, \ldots, n\}$ of size $n - f$. According to Definition 4, we have

$$\|\hat{\boldsymbol{g}} - \bar{\boldsymbol{g}}_{\mathcal{G}}\|^2 \leq \frac{\kappa}{n - f} \sum_{i \in \mathcal{S}} \|\boldsymbol{g}_i - \bar{\boldsymbol{g}}_{\mathcal{G}}\|^2 \leq \frac{\kappa}{n - f} \sum_{i \in \mathcal{S}} \max_{j \in \mathcal{G}} \|\boldsymbol{g}_i - \boldsymbol{g}_j\| \leq \kappa \max_{i,j \in \mathcal{G}} \|\boldsymbol{g}_i - \boldsymbol{g}_j\|^2 \quad (88)$$

We take take the square root of both sides of Inequality (88), then we have

$$\|\hat{\boldsymbol{g}} - \bar{\boldsymbol{g}}_{\mathcal{G}}\| \leq \sqrt{\kappa} \max_{i,j \in \mathcal{G}} \|\boldsymbol{g}_i - \boldsymbol{g}_j\| \quad (89)$$

Therefore, $\mathcal{A}$ is $(f, \lambda)$-resilient for $\lambda = \sqrt{\kappa}$. $\qquad \square$

Combining Proposition 4 with Proposition 1 and Proposition 2, the following corollaries are obvious.

**Corollary 3.** *Given any $(f, \kappa)$-robust AGR $\mathcal{A}$, if the set of honest gradients $\{\boldsymbol{g}_i \mid i \in \mathcal{H}\}$ is $(f, \gamma)$-skewed, then there exist Byzantine gradients $\{\boldsymbol{g}_i \mid i \in \mathcal{B}\}$ such that*

$$\mathbb{E}[\|\mathcal{A}(\boldsymbol{g}_1, \ldots, \boldsymbol{g}_n) - \bar{\boldsymbol{g}}\|^2] \geq \Omega(\frac{\gamma}{\kappa} \cdot \frac{f^2}{(n - f)^2} \cdot \rho_{\mathcal{S}}^2). \quad (90)$$

*where $\bar{\boldsymbol{g}} = \sum_{i \in \mathcal{H}} \boldsymbol{g}_i / (n - f)$ is the optimal gradient, $\rho_{\mathcal{S}}^2 = \mathbb{E}[\max_{i,j \in \mathcal{S}} \|\boldsymbol{g}_i - \boldsymbol{g}_j\|^2]$, $\mathcal{S}$ is the index set of the skewed honest gradients.*

**Corollary 4.** *Given any $(f, \kappa)$-robust AGR $\mathcal{A}$, $L$-smooth global loss function $\mathcal{L}$, and learning rate $\eta \leq 1/L$, if honest gradients $\{\boldsymbol{g}_i^t \mid i \in \mathcal{H}\}$ are $(f, \gamma)$-skewed for all $t = 0, \ldots, T - 1$, then there exists Byzantine gradients $\{\boldsymbol{g}_b^t \mid b \in \mathcal{B}, t = 0, \ldots, T - 1\}$ such that the global model parameter is bounded away from the global optimum $\boldsymbol{w}^*$:*

$$\mathbb{E}[\|\boldsymbol{w}^t - \boldsymbol{w}^*\|^2] \geq \Omega(\eta^2 (1 - L\eta)^2 \cdot \frac{\gamma}{\kappa} \cdot \frac{f^2}{(n - f)^2} \cdot \rho^2), \quad t = 1, \ldots, T, \quad (91)$$

*where $\boldsymbol{w}^t$ is the parameter of global model in the $t$-th communication round, $\boldsymbol{w}^*$ is the global optimum of global loss function $\mathcal{L}$, $\rho^2 = \min_{t=0, \ldots, T-1} \mathbb{E}[\max_{i,j \in \mathcal{S}^t} \|\boldsymbol{g}_i^t - \boldsymbol{g}_j^t\|^2]$, and $\mathcal{S}^t$ is the index set of the skewed honest gradients in $t$-th communication round.*

## C Algorithm of the Proposed STRKE Attack

## D Bisection Method to Solve Equation (19)

In this section, we present the bisection method used to solve Equation (19). We define $f(\cdot)$ as follows.

$$f(\alpha) = \max_{i \in \mathcal{S}} \|\bar{\boldsymbol{g}}_{\mathcal{S}} + \alpha \cdot \text{sign}(\bar{\boldsymbol{g}}_{\mathcal{S}}) \odot \boldsymbol{\sigma}_{\mathcal{S}} - \boldsymbol{g}_i\| - \max_{i,j \in \mathcal{S}} \|\boldsymbol{g}_i - \boldsymbol{g}_j\|, \quad \alpha \in [0, +\infty). \quad (92)$$

We can easily verify the following facts: 1. $f(0) \leq 0$, $f(\alpha) \to +\infty$ when $\alpha \to +\infty$; 2. $f(\cdot)$ is continuous; 3. $f(\cdot)$ has unique zero point in $[0, +\infty)$. Therefore, optimizing Equation (19) is equivalent to finding the zero point of $f(\cdot)$, which can be easily solved by bisection method in **??**.

---

**Algorithm 1** STRIKE Attack

---

**Require:** Honest gradients $\{\boldsymbol{g}_i \mid i \in \mathcal{H}\}$, hyperparameter $\nu > 0$ that controls attack strength (default $\nu = 1$)

$\boldsymbol{g}_{\mathrm{med}} \leftarrow$ Coordinate-wise median of $\{\boldsymbol{g}_i \mid i \in \mathcal{H}\}$      ▷ Stage 1: search for the skewed majority

$\boldsymbol{u}_{\mathrm{search}} \leftarrow \boldsymbol{g}_{\mathrm{med}} - \bar{\boldsymbol{g}}$

**for** $i \in \mathcal{H}$ **do**

     $p_i \leftarrow \langle \boldsymbol{g}_i, \boldsymbol{u}_{\mathrm{search}} / \|\boldsymbol{u}_{\mathrm{search}}\| \rangle$

**end for**

$\mathcal{S} \leftarrow$ Set of $n - f$ indices of honest gradients with the highest $p_i$

$\bar{\boldsymbol{g}}_{\mathcal{S}} \leftarrow \sum_{i \in \mathcal{S}} \boldsymbol{g}_i / (n - 2f)$      ▷ Stage 2: hide Byzantine gradients within the skewed majority

$\boldsymbol{\sigma}_{\mathcal{S}} \leftarrow$ Coordinate-wise standard deviation of $\{\boldsymbol{g}_i \mid i \in \mathcal{S}\}$

solve Equation (19) for $\alpha$

**for** $b \in \mathcal{B}$ **do**

     $\boldsymbol{g}_b \leftarrow \bar{\boldsymbol{g}}_{\mathcal{S}} + \nu \alpha \cdot \mathrm{sign}(\bar{\boldsymbol{g}}_{\mathcal{S}} - \bar{\boldsymbol{g}}) \odot \boldsymbol{\sigma}_{\mathcal{S}}$

**end for**

**return** Byzantine gradients $\{\boldsymbol{g}_b \mid g \in \mathcal{B}\}$

---

Theoretically, the final optimization problem of the proposed STRIKE attack in Eq. (19) can be effectively solved by a bisection algorithm as discussed in Appendix C. The computation cost is $\mathcal{O}(-\log \epsilon)$, where $\epsilon$ is the error of $\alpha$. In experiments, we perform bisection only 8 times and make the error of $\alpha$ within $1\%$. Empirically, the attack time for STRIKE is 12.11s (13.47s for MinMax, 13.35s for MinSum) on ImageNet-12.

In contrast, benign clients perform local updates to compute local gradients. The computation cost depends on the local data size, model architecture, batch size, number of local epochs, etc. In our setting, the average local update time is 15.14s on CIFAR-10, 11.76s on ImageNet-12 and 27.13s on FEMNIST. Tests are performed on a single A100 GPU.

## E    EXPERIMENTAL SETUPS AND ADDITIONAL EXPERIMENTS

### E.1    EXPERIMENTAL SETUPS

#### E.1.1    DATA DISTRIBUTION

For CIFAR-10 Krizhevsky et al. (2009) and ImageNet-12, we use Dirichlet distribution to generate non-IID data by following Yurochkin et al. (2019); Li et al. (2021a). For each class $c$, we sample $\boldsymbol{q}_c \sim \mathrm{Dir}_n(\beta)$ and allocate a $(\boldsymbol{q}_c)_i$ portion of training samples of class $c$ to client $i$. Here, $\mathrm{Dir}_n(\cdot)$ denotes the $n$-dimensional Dirichlet distribution, and $\beta > 0$ is a concentration parameter. We follow Li et al. (2021a) and set the number of clients $n = 50$ and the concentration parameter $\beta = 0.5$ as default.

For FEMNIST, the data is naturally partitioned into 3,597 clients based on the writer of the digit/character. Thus, the data distribution across different clients is naturally non-IID. For each client, we randomly sample a 0.9 portion of data as the training data and let the remaining 0.1 portion of data be the test data following Caldas et al. (2018).

#### E.1.2    HYPERPARAMETER SETTING OF BASELINES ATTACKS

The compared baseline attacks are: BitFlip (Allen-Zhu et al., 2020), LIE (Baruch et al., 2019), IPM (Xie et al., 2020), Min-Max (Shejwalkar & Houmansadr, 2021), Min-Sum (Shejwalkar & Houmansadr, 2021), and Mimic (Karimireddy et al., 2022). The hyperparameter setting of the above attacks is listed in the following table.

#### E.1.3    THE HYPERPARAMETER SETTING OF EVALUATED DEFENSES

The performance of our attack is evaluated on seven recent robust defenses: Multi-Krum Blanchard et al. (2017), Median Yin et al. (2018), RFA Pillutla et al. (2019), Aksel Boussetta et al. (2021), CClipKarimireddy et al. (2021) DnC Shejwalkar & Houmansadr (2021), and RBTM El-Mhamdi

Table 3: The hyperparameter setting of six baseline attacks. N/A represents there is no hyperparameter required for this attack.

| Attacks | Hyperparameters |
|---------|-----------------|
| BitFlip | N/A |
| LIE | $z = 1.5$ |
| IPM | $\varepsilon = 0.1$ |
| Min-Max | $\gamma_{\text{init}} = 10, \tau = 1 \times 10^{-5}, \nabla^p$: coordinate-wise standard deviation |
| Min-Sum | $\gamma_{\text{init}} = 10, \tau = 1 \times 10^{-5}, \nabla^p$: coordinate-wise standard deviation |
| Mimic | N/A |

et al. (2021). The hyperparameter setting of the above defenses is listed in the following table. we

Table 4: The hyperparameter setting of seven evaluated defenses. N/A represents there is no hyperparameter required for this defense.

| Defenses | Hyperparameters |
|----------|-----------------|
| Multi-Krum | N/A |
| Median | N/A |
| RFA | $T = 8$ |
| Aksel | N/A |
| CClip | $L = 1, \tau = 10$ |
| DnC | $c = 1, \text{niters} = 1, b = 1000$ |
| RBTM | N/A |

also consider a simple yet effective bucketing scheme (Karimireddy et al., 2022) that adapts existing defenses to the non-IID setting. We follow the original paper and set the bucket size to be $s = 2$.

### E.1.4 EVALUATION

We use top-1 accuracy, i.e., the proportion of correctly predicted testing samples to total testing samples, to evaluate the performance of global models. The *lower* the accuracy, the more effective the attack. We run each experiment five times and report the mean and standard deviation of the highest accuracy during the training process.

### E.1.5 COMPUTE

All experiments are run on the same machine with Intel E5-2665 CPU, 32GB RAM, and four GeForce GTX 1080Ti GPU.

### E.1.6 OTHER SETUPS

The number of Byzantine clients of all datasets is set to $f = 0.2 \cdot n$. We test STRIKE with $\nu \in \{0.25 \cdot i \mid i = 1, \ldots, 8\}$ and report the lowest test accuracy (highest attack effectiveness).

The hyperparameter setting for datasets FEMNIST (Caldas et al., 2018), CIFAR-10 (Krizhevsky et al., 2009) and ImageNet-12 (Russakovsky et al., 2015) are listed in below Table 5.

Table 5: Hyperparameter setting for FEMNIST, CIFAR-10 and ImageNet-12. # is the number sign. For example, # Communication rounds represents the number of communication rounds.

| Dataset | FEMNIST | CIFAR-10 | ImageNet-12 |
|---|---|---|---|
| Architecture | CNN (Caldas et al., 2018) | AlexNet (Krizhevsky et al., 2017) | ResNet-18 (He et al., 2016) |
| # Communication rounds | 800 | 200 | 200 |
| # Sampled Clients | 10 | 50 | 50 |
| # Local epochs | 1 | 1 | 1 |
| Optimizer | SGD | SGD | SGD |
| Batch size | 128 | 128 | 128 |
| Learning rate | 0.5 | 0.1 | 0.1 |
| Momentum | 0.5 | 0.9 | 0.9 |
| Weight decay | 0.0001 | 0.0001 | 0.0001 |
| Gradient clipping | Yes | Yes | Yes |
| Clipping norm | 2 | 2 | 2 |

### E.2    ADDITIONAL EXPERIMENTS

#### E.2.1    PERFORMANCE UNDER VARYING HYPERPARAMETER $\nu$

We study the influence of $\nu$ on ImageNet-12 dataset. We report the test accuracy under STRIKE attack with $\nu$ in $\{0.25 * i \mid i = 1, \ldots, 8\}$ against seven different defenses on ImageNet-12 in Figure 6. We also report the lowest test accuracy (best performance) of six baseline attacks introduced in Section 6.1 as a reference. Please note that a *lower* accuracy implies higher attack effectiveness.

As shown in the Figure 6, the performance of STRIKE is generally competitive with varying $\nu$. In most cases, simply setting $\nu = 1$ can beat other attacks (except for CClip, yet we observe that the performance is low enough to make the model useless). The impact of $\nu$ value is different for different robust AGRs: for Median and RFA, the accuracy is relatively stable under different $\nu$s; for CClip and Multi-Krum, the accuracy is lower with larger $\nu$s; for Aksel and DnC, the accuracy first decreases and then increases as $\nu$ increases.

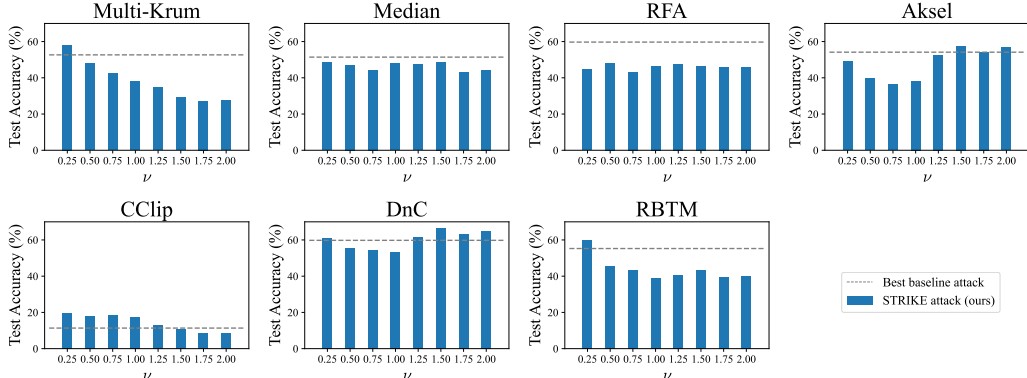

Figure 6: Accuracy under STRIKE attack with $\nu$ in $\{0.25 * i \mid i = 1, \ldots, 8\}$ against seven different defenses on ImageNet-12. The gray dashed line in each figure represents the lowest test accuracy (best performance) of six baseline attacks introduced in Section 6.1. We include it as a reference. The *lower* the accuracy, the more effective the attack. Other experimental setups align with the main experiment as introduced in Section 6.1.

### E.2.2 PERFORMANCE UNDER DIFFERENT NON-IID LEVELS

As shown in Table 1, DnC demonstrates the strongest robustness against various attacks on all datasets. Therefore, we fix the defense to be DnC in this experiment. As discussed in Appendix E.2.1, simply setting $\nu = 1$ yields satisfactory performance of our STRIKE attack. Thus, we fix $\nu = 1$ in this experiment. We vary Dirichlet concentration parameter $\beta$ in $\{0.1, 0.2, 0.5, 0.7, 0.9\}$ to study how our attack behaves under different non-IID levels. Lower $\beta$ implies a higher non-IID level. We additionally test the performance in the IID setting. Other setups align with the main experiment as introduced in Section 6.1. The results are posted in Figure 7 below.

As shown in Figure 7, the accuracy generally increases as $\beta$ decreases for all attacks. The accuracy under our STRIKE attack is consistently lower than all the baseline attacks. Besides, we also note that the accuracy gap between our STRIKE attack and other baseline attacks gets smaller when the non-IID level decreases. We hypothesize the reason is that gradient skew is milder as the non-IID level decreases, which aligns with our theoretical results in Propositions 1 and 2. Even in the IID setting, our STRIKE attack is competitive compared to other baselines.

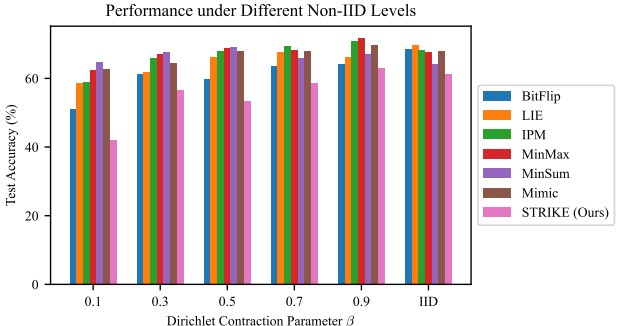

Figure 7: Accuracy under different attacks against DnC under different non-IID levels on ImageNet12. Lower $\beta$ implies a higher non-IID level. "IID" implies that the data is IID distributed. The *lower*, the better. Other setups align with the main experiment as introduced in Section 6.1.

### E.2.3 PERFORMANCE UNDER DIFFERENT BYZANTINE CLIENT RATIO

As shown in Table 1, DnC demonstrates the strongest robustness against various attacks on all datasets. Therefore, we fix the defense to be DnC in this experiment. As discussed in Appendix E.2.1, simply setting $\nu = 1$ yields satisfactory performance of our STRIKE attack. Thus, we fix $\nu = 1$ in this experiment. We vary the number of Byzantine clients $f$ in $\{5, 10, 15, 20\}$ and fix the total number of clients $n$ to be $50$. In this way, Byzantine client ratio $f/n$ varies in $\{0.1, 0.2, 0.3, 0.4\}$ to study how our attack behaves under different Byzantine client ratio. Other setups align with the main experiment as introduced in Section 6.1. The results are posted in Figure 8 below.

As shown in Figure 8, the accuracy generally decreases as $f/n$ increases for all attacks. The accuracy under our STRIKE attack is consistently lower than all the baseline attacks. The results suggest that all attacks are more effective when there are more Byzantine clients. Meanwhile, our attack is the most effective under different Byzantine client number.

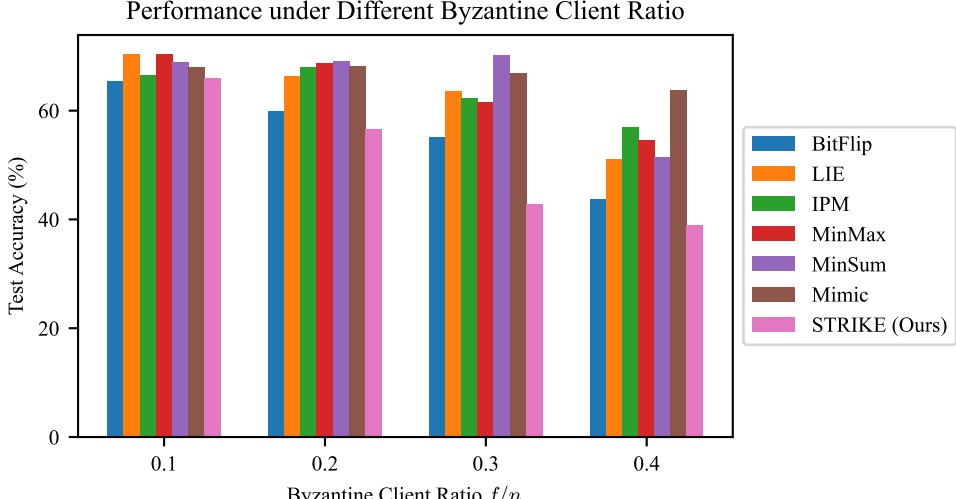

Figure 8: Accuracy under different attacks against DnC under different Byzantine client ratio on ImageNet12. The *lower*, the better. Other setups align with the main experiment as introduced in Section 6.1.

### E.2.4   PERFORMANCE UNDER DIFFERENT CLIENT NUMBER

As shown in Table 1, DnC demonstrates the strongest robustness against various attacks on all datasets. Therefore, we fix the defense to be DnC in this experiment. As discussed in Appendix E.2.1, simply setting $\nu = 1$ yields satisfactory performance of our STRIKE attack. Thus, we fix $\nu = 1$ in this experiment. We vary the number of total clients $n$ in $\{10, 30, 50, 70, 90\}$ and set the number of Byzantine clients $f = 0.2n$ accordingly. In this way, We can study how our attack behaves under different client number. Other setups align with the main experiment as introduced in Section 6.1. The results are posted in Figure 9 below.

As shown in Figure 9, the accuracy generally decreases as client number $n$ increases for all attacks. The accuracy under our STRIKE attack is consistently lower than all the baseline attacks under different client number.

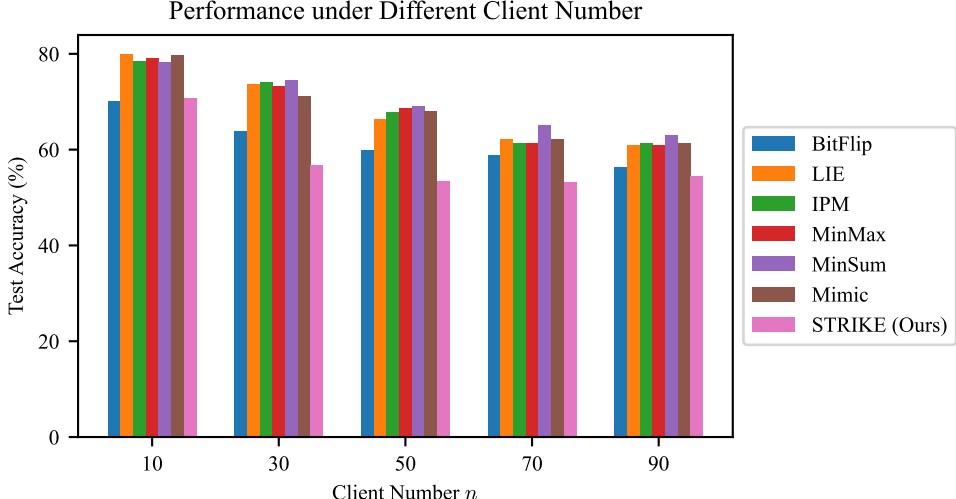

Figure 9: Accuracy under different attacks against DnC under different client number on ImageNet12. The *lower*, the better. Other setups align with the main experiment as introduced in Section 6.1.

