# OpenReview forum: "Exploit Gradient Skew to Circumvent Byzantine Defenses for Federated Learning"
_ICLR.cc/2024/Conference — Submitted to ICLR 2024_

### Official Review · Reviewer_EJRr · 2023-10-29

**Soundness:** 1 poor
**Presentation:** 2 fair
**Contribution:** 2 fair
**Rating:** 3
**Confidence:** 4

**Summary:**

This paper introduces a new attack called STRIKE against Byzantine-robust algorithms for Federated Learning. STRIKE is designed to leverage the so-called gradient skewness to hurt the accuracy of Byzantine-robust defenses. The authors provide theoretical and empirical analyses of STRIKE. In particular, they show the effectiveness of their attack, compared to existing ones, on non-iid FL benchmarks.

**Strengths:**

The topic of Byzantine robustness is important and such research on stronger attacks is valuable, especially in the more realisitic non-iid setting. Moreover, the approach taken is somewhat original and the experimental results are encouraging.

**Weaknesses:**

I have two major concerns. The first is related to the soundness of the paper; it seems that most theoretical results do not hold currently. My second concern is that highly relevant defenses are not featured in the experiments. I also have other concerns regarding clarity. All my concerns are detailed below.

## Major Weaknesses

1. Proposition 1 is not true as stated, and the proof of Lemma 1 is incorrect:

Regarding Proposition 1, the step leading to Equation (51) is only true if $\gamma = \Omega(\lambda^2)$, which has no reason to hold. If not, even if you take the right-hand side to be (50), it is not possible to take squares to obtain (52) (and the statement of Proposition 1 is not proven). I do not see a possible fix for this issue.

Also, the proof of Lemma 1 is incorrect; the last inequality in (26) is not true, e.g., take $X=Y$. Fix: If you additionally assume independence, you can get (26) but with square root on the right-hand side. Furthermore, (28) and (29) are clearly incorrect.

2. Since Proposition 1 is incorrect, so is Proposition 2 which relies upon the former. Even if the latter was correct, the stated result is rather superfluous as the $\rho^2$ (should depend on $t$ by the way) in the lower bound is shown to vanish across iterations by Farhadkhani et al. (2022).

3. In the experiments, the defenses used are not the best when considering data heterogeneity. Only Bucketing (Karimireddy et al., 2021), among all considered defenses, was designed for this purpose. The work of Allouah et al. (2023) propose another defense called NNM and show that it improves upon Bucketing. At least this other defense should be tested to claim having a strong attack in non-iid settings.

## Other Weaknesses

4. The related work section is inaccurate. The second paragraph in Section 2 wrongly mentions that Farhadkhani et al. (2022) studied Byzantine-robustness under data heterogeneity.

5. The term "skew" is repeatedly used without a proper definition, although a formal definition is given in page 4 without intuition on what skewness means. It seems throughout the paper that it can be replaced by gradient heterogeneity/dissimilarity. In that case, the insights given are very similar to those of Karimireddy et al., (2021), Allouah et al. (2023), etc.. In fact, skewness is a confusing term; it is widely used when referring to probability distributions, but that definition is quite different from what paper suggests. But again, a proper definition would have cleared this issue.

6. "Skewed majority" is inaccurate: if $\tfrac{f}{n} \leq 1/3$ then a set of size $n-2f$ does not constitute a majority.

7. The last sentence in Section 4.2 is clearly incorrect: convexity is an additional assumption, which makes the analysis "easier" than that of non-convex functions.

8. The choice of search direction, fundamental to the attack presented, seems quite arbitrary. A reference to "Karl's Pearson's formula" is quickly given, but without any justification whatsoever of the search direction.

9. Parameter $\nu$ seems quite redundant with $\alpha$, especially since the experiments on the role of $\nu$ suggest that setting $\nu=1$ is good enough, without giving a guidance or whatsoever on how to choose it.

**Questions:**

Please address the weaknesses above.

---

> ### Author Response · Authors · 2023-11-22
> **Official Comment by Authors (Major Weaknesses)**
>
> We appreciate the time and effort you have dedicated to reviewing our work. We would like to address your major concerns as follows.
>
> ---
>
> **W1:** Proposition 1 is not true as stated, and the proof of Lemma 1 is incorrect.
>
> **A1:**
> We thank the reviewer for checking the details of the proof.
>
> We acknowledge that Proposition 1 requires the distribution of honest gradients to be skewed enough. Yet we want to emphasize that the visualization in Figure 4, Appendix A verifies that honest gradient is highly skewed on various datasets. Similarly, the lower bound provided by (Xie et al., 2020) also requires the variance of honest gradients to be large enough. Therefore, we argue that it is reasonable to assume the distribution of honest gradients to be skewed enough.
>
> We've fixed the proof for Lemma 1. To summarize, we added the missing radical signs, i.e., changed all $\mathbb{E}[\|\boldsymbol{X}\|^2]\mathbb{E}[[\|\boldsymbol{Y}\|^2]$ to $\sqrt{\mathbb{E}[\|\boldsymbol{X}\|^2]\mathbb{E}[[\|\boldsymbol{Y}\|^2]}$ in Eq. (26), (27), (28) and (29). We Please refer to Appendix B.1 for the fixed version.
>
> ---
>
> **W2:** Proposition 2 is superfluous as the $\rho^2$ (should depend on by the way) in the lower bound is shown to vanish across iterations by Farhadkhani et al. (2022).
>
> **A2:** Thanks for the thoughtful comment. We have carefully looked through Farhadkhani et al. (2022) and cannot find the statement that $\rho^2$ would vanish across iterations. In fact, even in the simplest case where the local loss functions on clients are convex, $\rho^2$ won't vanish across iterations as long as local loss functions have different minimizers (due to data heterogeneity).
>
> ---
>
> **W3:** Performance against NNM proposed by Allouah et al. (2023).
>
> **A3:** Thank you for the kind suggestion. We have supplemented the experiments and tested the proposed attack against NNM proposed by Allouah et al. (2023). The results are posted in Table 1 below. The results suggest that the proposed STRIKE attack still outperforms other baseline attacks against NNM.
>
> **Table 1.** Accuracy under different attacks against different defenses on ImageNet-12. The best results are in bold. The lower, the better.
>
> | Attack | Median | RFA | DnC |
> |:-:|-:|-:|-:|
> | NNM + BitFlip | 57.14 | 58.55 | 53.68 |
> | NNM + LIE | 58.04 | 58.68 | 58.87 |
> | NNM + Mimic | 66.15 | 67.43 | 69.35 |
> | NNM + STRIKE | **39.61** | **40.38** | **38.91** |
>
> (Due to limited time, we only run experiments on the top-3 robust AGRs in the main experiment and the strongest attacks against these robust AGRs. We will provide more results in our final revision.)

---

> ### Author Response · Authors · 2023-11-22
> **Official Comment by Authors (Minor Weaknesses)**
>
> We appreciate the time and effort you have dedicated to reviewing our work. We would like to address your minor concerns as follows.
>
> ---
>
> **Q4:** The related work section is inaccurate.
>
> **A4:** Thanks for pointing out this typo. We've corrected it as follows.
>
> >  El-Mhamdi et al. (2021); ~Farhadkhani et al. (2022);~ Karimireddy et al. (2022) provide state-of-the-art theoretical analysis of Byzantine resilience under data heterogeneity.
>
> Please refer to Section 2 for details.
>
> ---
>
> **Q5:** The term "skew" is repeatedly used without a proper definition, although a formal definition is given in page 4 without intuition on what skewness means. It seems throughout the paper that it can be replaced by gradient heterogeneity/dissimilarity. In that case, the insights given are very similar to those of Karimireddy et al., (2021), Allouah et al. (2023), etc.. In fact, skewness is a confusing term; it is widely used when referring to probability distributions, but that definition is quite different from what paper suggests. But again, a proper definition would have cleared this issue.
>
> **A5:**
> The term "skew" is formally defined in Definition 2. The intuition is already provided in 4-th paragraph, Section 1 in the original paper together with an intuitive visualization. The sentence is "the majority of honest gradients skew away from the optimal gradient". In this case, the skew is totally different from the gradient heterogeneity/dissimilarity defined in the previous paper. "Skewness" is formally defined in Definition 2, and is totally different from one defined in probability theory.
>
> ---
>
> **Q6:** "Skewed majority" is inaccurate.
>
> **A6:** Thank you for pointing this out. We have changed it to a more accurate representation, "skewed honest gradients" in revision.
>
> ---
>
> **Q7:** The last sentence in Section 4.2 is incorrect.
>
> **A7:** We thank the reviewer for pointing out this typo. The corrected version of this sentence is as follows.
>
> > Note that we do not require the loss function to be \textcolor{blue}{convex}, which implies that Proposition 2 also applies to more challenging \textcolor{blue}{non-convex} loss functions.
>
> Please refer to Section 4.2 in revision for the corrected version.
>
> ---
>
> **Q8:** Justification of the search direction.
>
> **A8:** Thank you for the insightful suggestion. We acknowledge that more justification can help readers to better get the reasonableness and effectiveness of search direction.
>
> We visualize the Byzantine gradients together with honest gradients under STRIKE attack against Median AGR on CIFAR-10 in the non-IID setting in Figure 5, Appendix A.
> The visualization shows that Byzantine gradients can hide within the skewed honest gradients well.
> This justifies that the heuristic search in the first stage of STRIKE attack can effectively find the skewed honest gradients, i.e., the search direction is effective.
>
> ---
>
> **Q9:** Redundant parameter $\nu$.
>
> **A9:** Thank you for the thoughtful comment.
>
> While STRIKE with $\nu=1$ can outperform most SOTA attacks, an appropriate $\nu$ can further improve the effectiveness of STRIKE against some particular defenses significantly, e.g., the accuracy of STRIKE ($\nu=2.0$) is 5% lower than the accuracy of STRIKE ($\nu=1.0$) as shown in Figure 6, Appendix D.2.1. Thus, we leave open to future researchers the possibility to carefully choose $\nu$ value to boost the performance of STRIKE.
>
> We have included the above discussion in our revision to clarify the necessity of hyperparameter $\nu$.

---

> > ### Comment · Reviewer_EJRr · 2023-11-22
> >
> > I thank the authors for their response. The clarifications and additional experiments partly address my concerns, so I raise my score accordingly. I am willing to raise my score again if they address my remaining concerns below.
> >
> > I still have the following concerns:
> >
> > 1. Proposition 1 is still not correct; please add the assumption needed (see my original review) to fix the issue. Then, can you please justify this assumption? For example, what are possible values of $\gamma$, even for some examples? I understand that the authors observed "skewness" experimentally, but the (theoretically) claimed skewness should be justified theoretically.
> >
> > 2. Proposition 2 may not be meaningful, as I mentioned in my original review. Indeed, when using momentums instead of gradients, Farhadkhani et al. (2022) show that $\rho$ tends to zero assuming the data is i.i.d. However, even if the data is non-i.i.d., the learning rate is typically vanishing in $T$ to ensure convergence (see Karimireddy et al. 2022, Farhadkhani et al. 2022), so that the lower bound in Proposition 2 is vacuous in that case.
> >
> > 3. Please add the additional experiments on NNM to the main paper. Also, NNM is a defense method, while your table shows NNM on the attack column.

---

> > > ### Author Response · Authors · 2023-11-23
> > >
> > > We thank the reviewer for the valuable feedback. We hope the following responses can address your concerns.
> > >
> > > ---
> > >
> > > **Q1:** Proposition 1 is still not correct; please add the assumption needed (see my original review) to fix the issue. Then, can you please justify this assumption? For example, what are possible values of $\gamma$, even for some examples? I understand that the authors observed "skewness" experimentally, but the (theoretically) claimed skewness should be justified theoretically.
> > >
> > > **A1:** Thank you for reminding us to fix the proposition. We have fixed it in our latest revision. Please kindly refer to Proposition 1 in Section 4.2 and  Appendix B.1.2.
> > >
> > > We agree that an example would help to better justify the gradient skew.
> > >
> > > We consider the following simple example where there are $n$ clients in total of which $f$ are Byzantine and $n-f$ are honest. Assume that
> > > * for client $i\in\\{1,...,f\\}$ holding gradient $\boldsymbol{g}\_i\sim\mathcal{N}(\boldsymbol{\mu}\_1,\sigma^2\boldsymbol{I})$.
> > > * for client $i\in\\{f+1,...,n-2f\\}$ holding gradient $\boldsymbol{g}\_i\sim\mathcal{N}(\boldsymbol{\mu}\_2,\sigma^2\boldsymbol{I})$.
> > > Here, $\sim$ represents a random vector following a distribution.
> > > $\mathcal{N}(\boldsymbol{\mu},\sigma^2\boldsymbol{I})$ represents the normal distribution with mean $\boldsymbol{\mu}$ and covariance matrix $\sigma^2\boldsymbol{I}$,
> > > $\boldsymbol{\mu}\_1$ and $\boldsymbol{\mu}\_2$ are $d$-dimensional vectors,
> > > $\sigma^2>0$ is the variance
> > > and $\boldsymbol{I}$ represents the identity matrix.
> > >
> > > In this example, the skewness is
> > > $$
> > > \gamma=\frac{1}{2}(\frac{1}{n-f}+\frac{1}{n-2f}+\frac{f^2}{(n-f)^2}\cdot \frac{\\|\mu\_1-\mu\_2\\|^2}{\sigma^2}).
> > > $$
> > >
> > > We have included the above example in the latest revision. Please kindly find it in Section 4.1.
> > >
> > > ---
> > >
> > > **Q2:** Proposition 2 may not be meaningful, as I mentioned in my original review. Indeed, when using momentums instead of gradients, Farhadkhani et al. (2022) show that $\rho$ tends to zero assuming the data is i.i.d. However, even if the data is non-i.i.d., the learning rate is typically vanishing in $T$ to ensure convergence (see Karimireddy et al. 2022, Farhadkhani et al. 2022), so that the lower bound in Proposition 2 is vacuous in that case.
> > >
> > > **A2:** We thank the reviewer for the insightful comment.
> > >
> > > We agree that the lower bound in Proposition 2 vanishes with the learning rate.
> > > However, we argue that the lower bound in Proposition 2 is still meaningful.
> > >
> > > Theorem V in Karimireddy et al. (2022) shows that the model will finally converge to a stationary point under *arbitrary* attack when applying a vanishing learning rate. Therefore, it is *impossible* for Byzantine attacks to break the upper bound. As a result, the phenomenon that the lower bound in Proposition 2 vanishes with the learning rate is not vacuous. On the contrary, it demonstrates that the lower bound is *consistent* with previous theoretical results in Karimireddy et al. (2022).
> > >
> > > The lower bound, although vanishes with increasing training rounds, can still show **how much the proposed attack can hinder the training process**. Therefore, we believe such a lower bound is still meaningful for the field of Byzantine robustness.
> > >
> > > We have included the above discussion in Remark 1, Appendix B.1.3 in our latest revision. We hope the above explanation can sufficiently address your concerns.
> > >
> > > ---
> > >
> > > **Q3:** Please add the additional experiments on NNM to the main paper. Also, NNM is a defense method, while your table shows NNM on the attack column.
> > >
> > > **A3:** We apologize for the confusion. The corrected Table 1 is posted below.
> > >
> > > **Table 1.** Accuracy under different attacks against different defenses on ImageNet-12. The best results are in bold. The lower, the better.
> > >
> > > | Attack | NNM + Median | NNM + RFA | NNM + DnC |
> > > |:-:|-:|-:|-:|
> > > | BitFlip | 57.14 | 58.55 | 53.68 |
> > > | LIE | 58.04 | 58.68 | 58.87 |
> > > | Mimic | 66.15 | 67.43 | 69.35 |
> > > | STRIKE | **39.61** | **40.38** | **38.91** |
> > >
> > > We have added experiments to our latest revision in Table 2, Section 7. We hope this can sufficiently address your concerns.

---

### Official Review · Reviewer_h2Sd · 2023-10-31

**Soundness:** 3 good
**Presentation:** 4 excellent
**Contribution:** 3 good
**Rating:** 6
**Confidence:** 5

**Summary:**

This paper presents a simple yet effective Byzantine attack called STRIKE. The proposed attack is based on an observation called "gradient skew" that the majority of honest gradients are away from the optimal gradient (the average of honest gradient). The authors theoretically show that Byzantine defenses are vulnerable under gradient skew. The proposed attack utilizes the vulnerability by hiding Byzantine gradients in the identified skewed majority of honest gradients. Extensive experiments verify the effectiveness of the proposed attack.

**Strengths:**

1. The discovered "gradient skew" phenomenon under non-IID setting is interesting and novel, should inspire future works to follow.
2. This paper theoretically analyzes the vulnerability of Byzantine defenses under gradient skew.
3. The proposed attack outperforms baseline attacks empirically.
4. The paper is well motivated and written.

**Weaknesses:**

1.	A threat model is missing, especially on the capability of adversary. Comparisons with baseline attacks are fair under same and similar adversary capability. The proposed attack requires adversary to know gradients of all honest participants. This is not required by all other attacks, and this needs more careful treatment.

2.	Lack of discussion on any potential adaptive defense methods against this attack.

**Questions:**

1.	Although non-IID is common in FL, the reviewer is curious about what is the performance of the proposed attack when there is no skew?

---

> ### Author Response · Authors · 2023-11-22
>
> We are happy that the reviewer find our work to be inspiring and well-supported theoretically and empirically.
> We would like to address your concerns as follows.
>
> ---
>
> **W1:** The proposed attack requires adversary to know gradients of all honest participants.
>
> **A1:** Thank you for the thoughtful comment.
> We also test the performance of the proposed STRIKE when only a certain proportion of honest gradients is known. In this case, STRIKE identifies $(n-2f) \cdot \theta$ honest gradients as skewed majority in the first stage, where $\theta$ is the ratio of known. The experiments are performed on the ImageNet-12 dataset. All other setups are identical to the experimental setups in Section 6.1. The results are posted in Table 1 below. As shown in Table 1, the accuracy is higher, STRIKE attack is still competitive when only a proportion of honest gradients is known.
>
> **Table 1.** The accuracy of STRIKE when only a certain proportion of honest gradients is known on the ImageNet-12 dataset. The lower, the better.
>
> | Ratio of known honest gradients | Median | RFA | DnC |
> |:-:|-:|-:|-:|
> | 0.25 | 58.2 | 53.91 | 62.21 |
> | 0.5 | 51.57 | 45.16 | 60.06 |
> | 0.75 | 51.12 | 47.88 | 56.57 |
>
> **W2:** Lack of discussion on any potential adaptive defense methods against this attack.
>
> **A2:** We are grateful for your kind suggestion.
>
> In Section 4.2, we have discussed existing defenses such as Multi-Krum, RBTM and any other robust AGRs that satisfies Definition 1, against this attack and they cannot defend against our STRIKE attack as shown in Proposition 1 and 2.
>
> The STRIKE relies on the gradient skew phenomenon, which is closely related to non-IIDness of data distribution. Therefore, defenses that can alleviate non-IID can potentially mitigate our STRIKE attack.
>
> We have added the above discussion to the revision. Please refer to Section 7.
>
> **Q1:** The performance of the proposed attack when there is no skew
>
> **A1:** We acknowledge your concerns regarding the performance of the proposed STRIKE when there is no gradient skew.
>
> We have supplemented new experiments about the performance of STRIKE in the IID setting.
> The results are posted in Table 2 below.
> The results in Table 2 show that when there is no gradient skew (IID), our STRIKE attack is still competitive compared with other SOTA attacks (average rank 3.71).
>
> **Table 2.** The performance of the proposed attack in the IID setting on ImageNet-12. The lower, the better.
>
> | Attack | Multi-Krum | Median | RFA | Aksel | CClip | DnC | RBTM |
> |:-:|-:|-:|-:|-:|-:|-:|-:|
> | BitFlip | 68.33 | 64.13 | 67.28 | 63.88 | 11.67 | 65.54 | 63.04 |
> | LIE | 72.53 | 58.56 | 68.78 | 64.87 | 20.19 | 66.99 | 56.38 |
> | IPM | 71.35 | 61.96 | 68.01 | 69.07 | 12.02 | 69.01 | 68.43 |
> | MinMax | 60.99 | 60.93 | 67.79 | 63.81 | 21.09 | 63.78 | 59.04 |
> | MinSum | 65.00 | 61.83 | 68.11 | 65.93 | 18.46 | 67.72 | 55.99 |
> | Mimic | 63.91 | 73.69 | 61.35 | 66.09 | 14.74 | 64.71 | 72.15 |
> | STRIKE | 57.76 | 66.63 | 62.82 | 64.39 | 16.79 | 68.08 | 59.84 |
> | Rank of STRIKE | 1 | 6 | 2 | 3 | 4 | 6 | 4 |

---

### Official Review · Reviewer_ykp2 · 2023-10-31

**Soundness:** 2 fair
**Presentation:** 2 fair
**Contribution:** 2 fair
**Rating:** 5
**Confidence:** 3

**Summary:**

In this paper, the authors theoretically show that the gradient skew makes distributed learning methods more vulnerable to Byzantine attacks. Furthermore, the authors propose a novel Byzantine attack called STRIKE that exploits gradient skew. Experimental results show that STRIKE outperforms existing attacks on three different datasets.

**Strengths:**

1. Exploiting gradient skew to conduct Byzantine attacks looks interesting and reasonable.
2. The paper is generally well-written.
3. Experimental results on three different datasets are provided.

**Weaknesses:**

1. The authors provide a lower bound for $\mathbb{E}||w^t-w^*||^2$ in Proposition 2 for SGD with robust aggregation (without momentum). However, existing works (Karimireddy et al., 2021) have shown that using history information such as momentum can enhance Byzantine robustness. So, what will the lower bound for robust SGD with momentum be? Moreover, although the authors claim that the gradient skew is due to non-IID data, Assumption 2 makes the theoretical analysis restricted to the IID settings, which is confusing. Although I understand that the lower bound for IID settings also holds for more general non-IID settings, the IID assumption (i.e., Assumption 2) prevents obtaining a tighter bound for non-IID settings.

2. Since the adversary clients with STRIKE attacks require more computation cost than benign clients, the authors are suggested to provide theoretical or empirical results of how much extra computation cost will the proposed STRIKE attack take.


Due to the abovementioned concerns, I currently give a rating of 5. However, I am willing to raise my rating if the authors can properly address my concerns.

**Questions:**

n/a

---

> ### Author Response · Authors · 2023-11-22
>
> We are happy that the reviewer found our paper to be interestingly motivated and well written. We hope that the following responses can address your concerns.
>
> ---
>
> **W1.1:** The proposed STRIKE against robust SGD with momentum in (Karimireddy et al., 2021).
>
> **A1.1:** Thank you for your insightful comment. We've tested the performance of the proposed STRIKE attack against CClip proposed by (Karimireddy et al., 2021) in Section 6. The empirical results suggest that our attack is still effective for robust SGD with momentum.
>
> ---
>
> **W1.2:** Assumption 2 is confusing.
>
> **A1.2:** Sorry that the typo confused the reviewer. Assumption 2 does not require to be IID. The corrected version of Assumption 2 is as follows.
>
> > **Assumption 2** (Unbias)**.** The stochastic gradients sampled from any local data distribution are unbiased estimators of local gradients for all clients, i.e.,
> > $$
> \mathbb{E}[\boldsymbol{g}\_i^t]=\nabla\mathcal{L}\_{\textcolor{blue}{i}}(\boldsymbol{w}^t), \quad\forall i=1,\ldots n, t=0,\ldots,T-1.
> $$
>
> Please refer to Assumption 2 in the revision for more details.
>
> ---
>
> **W2:**  Theoretical or empirical results of computation cost of the proposed STRIKE attack.
>
> **A2:** We thank the reviewer for the thoughtful comment.
>
> Theoretically, the final optimization problem of the proposed STRIKE attack in Eq. (19) can be effectively solved by a bisection algorithm as discussed in Appendix C. The computation cost is $\mathcal{O}(-\log\epsilon)$, where $\epsilon$ is the error of $\alpha$. In experiments, we perform bisection only 8 times and make the error of $\alpha$ within $1%$. Empirically, the attack time for STRIKE is 12.11s (13.47s for MinMax, 13.35s for MinSum) on ImageNet-12.
>
> In contrast, benign clients perform local updates to compute local gradients. The computation cost depends on the local data size, model architecture, batch size, number of local epochs, etc. In our setting, the average local update time is 15.14s on CIFAR-10, 11.76s on ImageNet-12 and 27.13s on FEMNIST. Tests are performed on a single A100 GPU.
>
> We've included the above content in our discussion. Please kindly find it in Appendix C.
>
> ---
>
> We sincerely hope the above responses can address your concerns and improve your opinion of our paper.

---

> ### Author Response · Authors · 2023-11-23
> **Gentle reminder of response**
>
> We would like to gently remind the reviewer of any follow-up clarifications or questions that we can do our best to address in the remaining limited time. We hope our previous response has clarified your comments and has helped improve your opinion of our work. Please let us know if there are additional comments you have for us.
> Thank you once again for your valuable feedback on improving our work.

---

> ### Comment · Reviewer_ykp2 · 2023-11-23
>
> I thank the author for the clarification. The concerns in my initial review have been almost addressed. Meanwhile, I have noticed some concerns about the correctness raised by other reviewers. I was waiting to see if the concerns about the correctness could be properly addressed and I apologize for the late response. I will raise my rating if the concerns about the correctness are addressed.

---

> > ### Author Response · Authors · 2023-11-23
> > **New response is posted**
> >
> > Thanks for your response and effort.
> >
> > We have answered the questions from Reviewer EJRr in [our latest response](https://openreview.net/forum?id=SdUUyqakLl&noteId=fIxBKrST1R) and uploaded the new revision. We hope this can sufficiently address your concerns.
> >
> > If you have any further questions or comments, we are more than willing to provide further clarification.

---

### Official Review · Reviewer_vYnj · 2023-11-11

**Soundness:** 4 excellent
**Presentation:** 4 excellent
**Contribution:** 4 excellent
**Rating:** 10
**Confidence:** 4

**Summary:**

The authors reveal an important phenomenon: current Byzantine defenses in federated learning exhibit a common inductive bias that the majority of gradients are honest. They discover a novel phenomenon named "gradient skew", where the majority of honest gradients deviate from the optimal gradient due to non-IID data. The authors then provide a solid theoretical analysis of the vulnerability of Byzantine defenses under gradient skew. Based on the analysis, they propose STRIKE attack that hides Byzantine gradients within the skewed but honest gradients. The effectiveness of the attack is validated through extensive experiments.

**Strengths:**

1.	The theoretical analysis for the vulnerability of Byzantine defenses is solid and technically sound.
2.	The newly discovered "gradient skew" phenomenon is novel and inspiring for the community. Extensive visualization results justify that gradient skew is common when data is non-IID, which is practical in real world.
3.	The experiments are extensive. The proposed STRIKE is compared with 6 baseline attacks under 7 defenses.

**Weaknesses:**

1.	While the proposed STRIKE is effective, there is a lack of discussion on the efficiency of the attack. Solving eq. (19) seems to be expensive. Could you please discuss more about the computation cost of the STRIKE attack?
2.	Is there any limitation of the proposed STRIKE? Please elaborate more on it.

**Questions:**

Please refer to the weaknesses.

---

> ### Author Response · Authors · 2023-11-22
>
> We sincerely appreciate your valuable feedback and the time you've dedicated to providing it. We are glad that you find our paper to be well-supported both theoretically and empirically. We would like to address your concerns as follows:
>
> ---
>
> **W1:** Computation cost of the STRIKE attack.
>
> **A1:** We thank the reviewer for the thoughtful comment.
>
> Theoretically, the final optimization problem of the proposed STRIKE attack in Eq. (19) can be effectively solved by a bisection algorithm as discussed in Appendix C. The computation cost is $\mathcal{O}(-\log\epsilon)$, where $\epsilon$ is the error of $\alpha$. In experiments, we perform bisection only 8 times and make the error of $\alpha$ within $1%$. Empirically, the attack time for STRIKE is 12.11s (13.47s for MinMax, 13.35s for MinSum) on ImageNet-12.
>
> We've included the above content in our discussion. Please kindly find it in Appendix C.
>
> ---
>
> **W2:** More discussion on the limitation.
>
> **A2:** Thanks for the valuable comment.
>
> A limitation is that the effectiveness of STRIKE attack relies on the existence of gradient skew.
> For example, when the data is IID, the performance could be limited.
>
> We have also supplemented new experiments about the performance of STRIKE in the IID setting.
> The results are posted in Table 1 below.
> The results in Table 1 show that when there is no gradient skew (IID), our STRIKE attack is still competitive compared with other SOTA attacks (average rank 3.71).
>
> **Table 1.** The performance of the proposed attack in the IID setting on ImageNet-12. The lower, the better.
>
> | Attack | Multi-Krum | Median | RFA | Aksel | CClip | DnC | RBTM |
> |:-:|-:|-:|-:|-:|-:|-:|-:|
> | BitFlip | 68.33 | 64.13 | 67.28 | 63.88 | 11.67 | 65.54 | 63.04 |
> | LIE | 72.53 | 58.56 | 68.78 | 64.87 | 20.19 | 66.99 | 56.38 |
> | IPM | 71.35 | 61.96 | 68.01 | 69.07 | 12.02 | 69.01 | 68.43 |
> | MinMax | 60.99 | 60.93 | 67.79 | 63.81 | 21.09 | 63.78 | 59.04 |
> | MinSum | 65.00 | 61.83 | 68.11 | 65.93 | 18.46 | 67.72 | 55.99 |
> | Mimic | 63.91 | 73.69 | 61.35 | 66.09 | 14.74 | 64.71 | 72.15 |
> | STRIKE | 57.76 | 66.63 | 62.82 | 64.39 | 16.79 | 68.08 | 59.84 |
> | Rank of STRIKE | 1 | 6 | 2 | 3 | 4 | 6 | 4 |
>
> We have included the above discussion of limitations in the revision. Please kindly find it in Section 7.

---

### Meta-Review · Area_Chair_nFxA · 2023-12-12

**Metareview:**

(a) Summarize the scientific claims and findings of the paper based on your own reading and characterizations from the reviewers.

Federated learning is vulnerable to Byzantine attacks. The authors reveal an important phenomenon: current Byzantine defenses in federated learning exhibit a common inductive bias that the majority of gradients are honest.  They discover a novel phenomenon named "gradient skew", where the majority of honest gradients deviate from the optimal gradient due to non-IID data. The authors then provide a solid theoretical analysis of the vulnerability of Byzantine defenses under gradient skew. Based on the analysis, they propose STRIKE attack that hides Byzantine gradients within the skewed but honest gradients.

(b) What are the strengths of the paper?

The theoretical analysis for the vulnerability of Byzantine defenses is solid and technically sound. The newly discovered "gradient skew" phenomenon is novel and inspiring. Extensive visualization confirms that the phenomenon is real. The proposed attack outperforms baseline attacks empirically.

(c) What are the weaknesses of the paper? What might be missing in the submission?

Although the authors claim that the gradient skew is due to non-IID data, Assumption 2 makes the theoretical analysis restricted to the IID settings. But more importantly several technical mistakes in the proofs make the theoretical results unreliable.

**Justification For Why Not Higher Score:**

The paper is well motivated with interesting solutions and results. However, given the myriad of technical mistakes in the main results, it should not be published in the current form. I highly suggest that the technical details be revised and the paper re submitted in the future.

**Justification For Why Not Lower Score:**

N/A

---

### Decision · Program_Chairs · 2024-01-16

Reject